
# Unraveling hydrometeor mixtures in polarimetric radar measurements

Nikola Besic[1,2], Josué Gehring[1], Christophe Praz[1], Jordi Figueras i Ventura[2], Jacopo Grazioli[2], Marco Gabella[2], Urs Germann[2], and Alexis Berne[1]

[1]Environmental Remote Sensing Laboratory (LTE), École Polytechnique Fédérale de Lausanne (EPFL), Lausanne, Switzerland
[2]Radar, Satellite, Nowcasting Department (MDR), MeteoSwiss, Locarno-Monti, Switzerland

**Correspondence:** Nikola Besic (nikola.besic@epfl.ch)

**Abstract.**

Radar-based hydrometeor classification typically comes down to determining the dominant type of hydrometeor populating a radar sampling volume. In this paper we address the subsequent problem of inferring the secondary hydrometeor types present in a volume - the issue of hydrometeor de-mixing. The presented study relies on the semi-supervised hydrometeor classification proposed by Besic et al. (2016), but nevertheless results in solutions and conclusions of a more general character and applicability. In the first part, dominantly marked by synthesis, a bin-based de-mixing approach is proposed, inspired by the conventional coherent and linear decomposition methods widely employed across different remote sensing disciplines. Intrinsically related to the concept of entropy, introduced in the context of the radar hydrometeor classification in Besic et al. (2016), the proposed method, based on the hypothesis of coherency in backscattering, estimates the proportions of different hydrometeor types in a given radar sampling volume, without considering the wider spatial context. Plausibility and performances of the method are evaluated using C and X band radar measurements, compared with hydrometeor properties derived from a Multi Angle Snowflake Camera instrument. In the second part, we examine the influence of the potential incoherency in the backscattering from different hydrometeors populating a radar sampling volume. This part, consists of adapting and testing the techniques commonly used in conventional incoherent decomposition methods to the context of weather radar polarimetry. The impact of the incoherence is found to be limited, justifying the hypothesis of coherency even in a case of mixed volumes, and confirming the applicability of the proposed bin-based approach.

## 1 Introduction

Precipitation, and in particular snowfall, often occurs as a mixture of several different hydrometeor types (Bringi and Chandrasekar, 2001). In the context of radar meteorology, we define hydrometeor mixture as a radar sampling volume populated by hydrometeors of different types. As such, it is more frequent in the regions of the atmosphere experiencing marked transitions between different hydrometeor types, due to specific microphysical processes like on-set of aggregation, riming, and melting. The probability of its occurrence increases with the distance from the radar, given the increase of the radar sampling volume. This type of sub-grid heterogeneity is not taken into account by classical hydrometeor classification techniques, which assign





a single label to the entire radar sampling volume. Therefore, a hydrometeor classification should ideally be complemented by a de-mixing step which gives an insight within a radar sampling volume when that proves to be relevant. More precisely, the term de-mixing refers to the attempt to systematically identify and quantify the presence of mixtures of different hydrometeor types in the radar sampling volume, as it has been specifically done for the rain-hail mixed precipitation (Balakrishnan and

Zrnic, 1990).

The hydrometeor classification is a very popular topic in weather radar community, particularly since the dual polarization radar became a widely used technology (Bringi et al., 2007). In its dominant, supervised form, the methods have been initially based on Boolean logic decision trees (Straka and Zrnic, 1993), before being replaced by a strong tendency to rely on fuzzy logic routine (Vivekanandan et al., 1999). The hypotheses about the microstructure and the microphysics of the precipitation are

used to simulate the polarimetric signatures of different hydrometeor types, which are then either directly (Dolan and Rutledge, 2009), or reinforced by some empirical knowledge (Al-Sakka et al., 2013), employed in defining fuzzy logic membership functions. Unlike this, the unsupervised method proposed by Grazioli et al. (2015b) has rather confidence in the acquired radar data, and distinguishes between different hydrometeor types by clustering the polarimetric radar observations.

In and effort to find a combination of these two conceptually opposed ideas, Besic et al. (2016) proposed a semi-supervised

method, which clusters the data by including the hypotheses about the microstructure and the microphysics as a constraint. The method provides a set of centroids in a space formed by four polarimetric variables and a precipitation phase indicator, reducing the classification problem to the simple computation of Euclidean distances. This simplicity made it possible for the method to be operationally implemented within the processing chain of the MeteoSwiss radar network (Germann et al., 2015), which allows to survey and in a way continuously verify and evaluate the performances of the classification.

Aside from assigning a label with the hydrometeor type to every volume, Besic et al. (2016) proposed a complementary measure of entropy, which gives an estimate of the classification uncertainty, and is therefore a potential indicator of hydrometeor mixtures. In the present study, the entropy parameter is appropriately parameterized using a synthetic dataset and serves as basis for a proposed bin-based de-mixing method (Fig. 1a). This approach does not consider the content of the surrounding volumes, but only the polarimetric parameters integrated over the volume of interest. The method is inspired by conventional

model-based decompositions of Synthetic Aperture Radar (SAR) data (Massonnet and Souyris, 2008) and linear unmixing of hyperspectral data (Bioucas-Dias et al., 2013). Its efficiency is assessed throughout the article by means of appropriate performance analyses. The latter includes simultaneous employment of the mobile MXPol X-band radar and MeteoSwiss C-band radars, as well as the collocated Multi Angle Snowflake Camera (MASC).

The article also investigates the potential impact of residual incoherency in weather radar measurements. This effect can be

presumed to be likely in the case of hydrometeorly mixed radar sampling volumes, despite the conventional pulse averaging which should deal with the influence of the spatial incoherence. Namely, more emphasized random interferences between the backscattered signals from different hydrometeors populating a radar sampling volume can be expected in case of a more pronounced heterogeneity among particles in a volume. The study is done through the neighborhood-based analysis, which is conducted by introducing the Blind Source Separation (BSS) techniques: Principal Component Analysis (PCA) and Inde-

pendent Component Analysis (ICA) into the weather radar data processing (Fig. 1b). That is to say, this part comes down to



the employment of the BSS techniques over a region of hydrometeor mixtures with the aim of asserting the influence of the residual incoherency on the weather radar measurements, and by doing so, verifying the applicability of the proposed bin-based approach, intrinsically related to the hypothesis of coherent backscattering.

[Figure 1 about here.]

The article is organized as follows: In Section 2 we briefly introduce the polarimetric framework we rely upon in discriminating between different hydrometeor types as well as the more general concept of entropy in the context of remote sensing. In Section 3 we introduce and elaborate the bin-based approach, after describing the entropy parametrization. This section also contains the performance analyses subsection, focused on the bin-based approach. In Section 4 we present the analysis, based on the conventional statistical techniques, and dedicated to the incoherency in the mixed radar sampling volumes. The

subsequent evaluation subsection illustrates the impact of incoherency in the context of weather radar de-mixing. Section 5 concludes the article with a discussion and provides a series of perspectives for the presented work.

## 2    Polarimetric framework and the concept of entropy

The main objective of this paper could be summarized as drawing a parallel between the specific polarimetric framework of weather radar and the paradigm of decomposition/unmixing commonly used in SAR and hyperspectral remote sensing. The

motivation for relying on the experience of SAR and hyperspectral communities comes from the demonstrated pertinence and utility of decomposition/unmixing in the data interpretation. This link we want to elaborate on with the aim of de-mixing a weather radar sampling volume, is constructed around the common variable - entropy.

### 2.1    Polarimetric framework

The radar variables we rely upon to discriminate between different hydrometeor types are: the reflectivity factor at horizontal

polarization ($Z_H$), the differential reflectivity ($Z_{DR}$), the specific differential phase shift ($K_{dp}$), and the co-polar correlation ($\rho_{hv}$). The hydrometeor classification (Besic et al., 2016) and the bin-based de-mixing approach proposed in this article also consider a phase indicator (Ind), that can be derived from radar data in stratiform cases or from external information like ground observations or model simulations. The latter has values: Ind $\approx -1$ for the liquid phase, Ind $\approx 0$ for the mixed phase and Ind $\approx 1$ for the solid phase (the approximative character is due to the employed sigmoid transformation of the relative

altitude with respect to $0°$ isotherm).

Due to the skewness and the leptokurticity of their distributions, $K_{dp}$ and $\rho_{hv}$ undergo the following logarithmic transformations: $K'_{dp} = 10\log{(K_{dp} + 0.6)}$ and $\rho'_{hv} = 10\log{(1 - \rho_{hv})}$. Further on, all four radar variables are linearly scaled i.e. min-max transformed ($[\cdot]^{\text{scaled}}$) to the $[-1, 1]$ range in the following limits: $Z_H$: -10 — 60 dBZ, $Z_{DR}$: -1.5 — 5 dB, $K'_{dp}$: -10 — 7 and $\rho'_{hv}$: -50 — -5.23.

[Figure 2 about here.]





Therefore, each radar sampling volume is characterized by a five elements weather radar target vector:

$$\mathbf{k} = \begin{bmatrix} x_1 \\ x_2 \\ x_3 \\ x_4 \\ x_5 \end{bmatrix} = \begin{bmatrix} Z_H^{\text{scaled}} \\ Z_{DR}^{\text{scaled}} \\ K_{dp}^{\prime\,\text{scaled}} \\ \rho_{hv}^{\prime\,\text{scaled}} \\ \text{Ind} \end{bmatrix}, \tag{1}$$

and can be represented as a point in a five-dimensional space formed by the five introduced parameters (Fig. 2a). The same space contains nine particular points i.e. centroids ($\mathbf{k}_c$), which represent the nine hydrometeor classes and the classification itself comes down to the calculation of the Euclidean distance between the centroids and the observations (Fig. 2b):

$$d = ||\mathbf{k}_c - \mathbf{k}||_2, \tag{2}$$

where $|| \cdot ||_2$ is the $\ell_2$ norm.

## 2.2 The concept of entropy

The concept of the entropy ($H$) has been introduced in the radar polarimetry by Cloude and Pottier (1997), via the decomposition theory. It serves as an indicator of the usefulness of the polarimetry, demonstrating simultaneously important discriminating capabilities in the context of the target classification. By equalizing the proportion of $n$ different components from the polarimetric decomposition to the probability of their occurrence ($p_i$), we obtain the entropy parameter which converges towards one if we do not have a clearly dominant component (a "total" mixture), or towards zero if we identify a clearly dominant component.

The min-entropy, being the minimum of Rényi's entropies (Rényi, 1960), and the originally proposed version of entropy estimator in Besic et al. (2016), is substituted here by the Shannon entropy estimator for the purpose of coherence with the conventional usage of the parameter in the remote sensing community:

$$H = -\sum_{i=1}^{n} p_i \log_n p_i, \tag{3}$$

having values in the range $[0, 1]$.

The estimation of probabilities $p_i$, from now on occasionally referred to as proportions, is the focal point of the first part of this paper i.e. the bin-based approach, proposed in the following section. A version of entropy, a bit closer to its original meaning in the radar polarimetry (and therefore named $H_{CP}$ - after Cloude and Pottier), with a slightly different nature of $p_i$, is used in the second part of the paper, studying the influence of potential incoherency in backscattering in weather radar measurements.





## 3  Bin-based approach

The bin-based approach considers one volume at a time, without taking the neighboring spatial context into account. The proposed method is inspired by the SAR coherent polarimetric decomposition on elementary processes (e.g. Pauli and Krogager, Massonnet and Souyris (2008)) and the hyperspectral linear unmixing (Bioucas-Dias et al., 2012), and adapted to the context

of the semi-supervised hydrometeor classification (Besic et al., 2016). Like its SAR role model, this method assumes the coherent summing of backscattering responses of different hydrometeors populating a radar sampling volume. This hypothesis is supported by the conventional averaging of backscattering information over a number of successive radar pulses, which aims to remove the influence of random spatial interferences.

Namely, the coherent decomposition of POLSAR data relies on the first-order statistics and represents the scattering matrix

of a target ($S$) as a coherent sum of scattering matrix of elementary interactions (respectively, odd-bounce and double bounce with two different orientations):

$$[\mathbf{S}] = \begin{bmatrix} S_{HH} & S_{HV} \\ S_{HV} & S_{VV} \end{bmatrix} = \frac{S_{HH} + S_{VV}}{2} \begin{bmatrix} 1 & 0 \\ 0 & 1 \end{bmatrix} +$$
$$+ \frac{S_{HH} - S_{VV}}{2} \begin{bmatrix} 1 & 0 \\ 0 & -1 \end{bmatrix} + S_{HV} \begin{bmatrix} 0 & 1 \\ 1 & 0 \end{bmatrix}, \tag{4}$$

with $S_{XX}$ being the ratio of scattered and incident electrical field for different combinations of horizontal ($X = H$) and vertical

($X = V$) polarizations. By comparing this to our polarimetric framework introduced in Sec. 2.1, we can deduce that our standard mechanisms i.e. elementary interactions would correspond to different hydrometeor classes. The important difference which prevents us to apply the equivalent formalism, even under the assumption of the total coherence, is the different physical nature of the elements of the weather radar target vector (Eq. 1), which contains much less geometrical information than the scattering matrix. Therefore, the starting point of the bin-based de-mixing endeavor is to establish hydrometeor classes as

elementary processes and to find a reliable, alternative way to sum them up, which evades the lack of geometrical information.

[Figure 3 about here.]

The hyperspectral linear unmixing problem appears more analogous to our polarimetric framework (Fig. 3). Namely, the target of the hyperspectral vector $\mathbf{y}$ contains the reflectance in $n$ frequency bands and under the hypothesis of linearity can be represented as:

$$\mathbf{y} = \sum_{i=1}^{p} \alpha_i \mathbf{m}_i, \tag{5}$$

where $\mathbf{m}_i$ is the vector of the $i$th "pure" material i.e. endmember, while $\alpha_i$ would be its corresponding proportion. In the simplest of all cases, where we actually have the pure materials (e.g. types of ground or crop) among the observations, we are





dealing with the so-called pure-pixel unmixing, where the method is reduced to the optimization problem:

$$\min_{\mathbf{M},\mathbf{A}} ||\mathbf{Y} - \mathbf{MA}||_F$$

subject to: $\mathbf{A} \geq \mathbf{0}$, $\mathbf{I}_p^T \mathbf{A} = \mathbf{I}_n$, \hfill (6)

with $\mathbf{Y}$ being the matrix of observations, $\mathbf{A}$ the matrix containing the proportions of every endmember in every observation,

and $\mathbf{M}$ the matrix containing endmembers. Matrices $\mathbf{I}$ and $\mathbf{I}^T$ are respectively, the identity matrix and its transposed version. The problem can be represented geometrically as the estimation of a simplex around the observations, as illustrated in Fig. 3.

By comparing Fig. 2a and Fig. 3 one can notice the intuitive similarity of the problem. However, the major obstacle in applying the very efficient paradigm of the spectral unmixing to the particular context of hydrometeor de-mixing is the fact that our centroids do not really represent the endmembers in the five-dimensional space. Namely, the target vectors corresponding

to centroids are supposed to be the representative examples of different classes and they are thus not represented by the extreme values of our polarimetric parameters. Nevertheless, basing the estimates of proportions on the distance in the space populated by the standard mechanisms and measurements is a reasonable way to sum up the standard mechanisms.

Now that we have identified our centroids as standard mechanisms (analogy with PolSAR) and have found a mode for their addition via the distances in the Euclidean space of the classification, we have to adapt the latter to the particular nature of

the former. Namely, our centroids do not form a regular structure (e.g. cube), where under the assumption of coherence and linearity, the distance of a measurement with respect to the standard mechanism could be directly interpreted as the probability i.e. as the proportion of the given standard mechanism. They are rather non-uniformly distributed in the five-dimensional space. In order to deal with this, we adopted the varying slope exponential transformation of the distance between the measurement and the $i$th centroid ($d_i$) to the probability ($p_i$) i.e. the proportion of the hydrometeor class $i$ depicted by the centroid in the

measurement. The exponential function is chosen in order to account for the uncertainty around the centroid ($p$ has higher values in the vicinity of the centroid), while the varying slope ($t_i$) accounts for the irregular distribution of the centroids in the classification space:

$$p_i = e^{-t_i d_i}, \; i = 1, ...8 \tag{9}$$ \hfill (7)

Namely, $t_i$ depends on the assigned classification label, which is basically the nearest centroid to the measurement ($c_i$). The

idea here is for a probability to drop to a threshold value $p_t$, at the distance corresponding to the separation between $c_i$ and $c_{closest}$ - the closest centroid to the one determining the label (Fig. 4):

$$p_t \approx e^{-t_i d(\mathbf{c}_i, \mathbf{c}_{closest})}, \hfill (8)$$

which results in:

$$t_i = \frac{\ln \frac{1}{p_t}}{d(\mathbf{c}_i, \mathbf{c}_{closest})}. \hfill (9)$$

[Figure 4 about here.]



The threshold value of probability ($p_t$) is determined using a synthetic dataset, created by linearly mixing different pairs of hydrometeor classes in equal proportions (an example in Fig. 5). Each "pure" hydrometeor box contains 900 synthetic realizations of hydrometeors varying uniformly in the very restricted interval around the values of the corresponding centroid ($-1\%$ to $1\%$, in order to emphasize the hypothesis of the "pureness"). Boxes of hydrometeor mixtures contain different versions

of mostly plausible mixtures obtained by linearly combining in equal shares polarimetric parameters of two hydrometeor types involved, following assumptions of coherence and linearity. Each of these boxes contains again 900 synthetic realizations, where every realization represents a presumingly equiprobable mixture. We opted for this "quadratic" organization of boxes, in order to indicate the limitations of the proposed approach as well, aside from obviously emphasizing the plausible mixtures.

[Figure 5 about here.]

Applying the hydrometeor classification on the synthetic dataset results in an expectedly proper recognition of pure hydrometeor boxes (Fig. 6). The mixed hydrometeor boxes are identified either as an ensemble of both hydrometeor classes involved in the mixing, an ensemble of only one of the classes involved in the mixing, or an ensemble of the classes which are not presumed to be involved in the mixing. The latter case, which mostly relates to the less plausible mixtures, represents the sort of limitation of the method (primarily boxes marked with the darkest shade of gray in Fig. 5a). Though, the co-existence of

two classes with very distinct polarimetric properties (particularly in terms of $Z_H$) is indeed not physically very probable. However, in these critical cases, the presence of the identified class which was not involved in the mixing process, can still be justified, e.g. the mixture of melting snow and melting hail contains some rain, or the mixture of ice hail and vertical ice contains aggregates and rimed particles.

The parametrized entropy, obtained by substituting $p_i$ in Eq. 3 with the one from Eq. 7 shows indeed very low values in case

of "pure" hydrometeors (Fig. 6). The synthetic hydrometeor mixtures are, on the other side, characterized by higher entropy values. This is true even for the less plausible mixtures mentioned in the previous paragraph, where the bin-based mixing approach reaches its limitations.

[Figure 6 about here.]

The utility of the proposed parametrization of entropy, which is based on the estimated proportions of different hydrometeors

involved in a mixture can be adequately illustrated using the exemplary data introduced in Fig. 2. In Fig. 7 we show the comparison between the original and the parametrized version of entropy values for the considered data samples. The increased values of entropy between the "mixable" neighbor centroids reflect clearly the results obtained with synthetic datasets. The limitation of the method can be also more clearly conceived here, because the mixtures of extreme points (e.g. vertical ice and ice hail) unavoidably finish close to the centroids in between.

[Figure 7 about here.]



## 3.1 Performance analyses

The performance of the introduced bin-based method is analyzed hereby in three respective stages, very characteristic for the validation of techniques related to the hydrometeor classification: spatial plausibility, comparison between two radars and comparison between a radar and a ground based instrument.

### 3.1.1 Spatial plausibility

In Fig. 8a1 we show an example of the classification applied on the MXPol X-band radar data acquired at the Dumont-d'Urville base, on the coast of Antarctica, during the APRES3 campaign (Grazioli et al., 2017). Given the systematically low temperatures at the ground level, in the illustrated range height indicator (RHI), only solid phase hydrometeors are identified. By analyzing the estimation of entropy, we can notice a rise in the entropy characterizing bordering regions between aggregation and riming. The high probability of mixing and misclassification of aggregates and rimed particles is due to the similarity in their polarimetric signatures and their spatial co-occurrence. Further more, in the context of meteorology, aggregates and rimed particles are not so distinct classes (a frequent phenomenon, riming of aggregates, is identified as rimed ice particle by the employed classification). All this calls for particular attention to this challenging and relevant de-mixing problem in the performance analyses. This is also sustained by our special interest in the riming identification due to its role in understanding better the orographic precipitation mechanisms (Grazioli et al., 2015a; Houze and Medina, 2005).

The results of the bin-based de-mixing, illustrated in Fig. 8b1 do not indicate the presence of hydrometeors other than the ones identified as dominant in Fig. 8a1. However, they indicate the significant percentage of aggregates in the volumes labeled as rimed ice particles and vice versa, as could be intuitively expected from the entropy estimate. The first stage of the performance analyses would be exactly the spatial continuity of this observation i.e. the fact that the percentage of rimed particles decreases progressively as we move away from the region labeled as rimed particles, the same being true for aggregates. No matter how simple this can seem, given that we deal with a bin-based method, which does not consider at all the spatial context, the observed spatial continuity can be indeed considered as the very first positive indicator of performances.

Fig. 8a2 depicts an example of the classification of a hail cell and its surrounding, observed by the MXPol X-band radar during the HyMeX campaign (Ducrocq et al., 2014; Bousquet et al., 2015). The results of the bin-based demixing (Fig. 8b2) show again the smooth and plausible transition between the lower part of the hail cell and the surrounding rain, with the borders of the cell (as defined by the classification) representing the intense mixing. The same situation is noted in the upper part of the cell, where hail mixes with the encircling rimed ice particles.

[Figure 8 about here.]

### 3.1.2 Inter-radar comparison

The following two stages of the performance verification are related to the measurement campaign organized in the Swiss canton of Valais, from November 2016 to April 2017. The campaign was based on the careful collocation of different instruments, depicted in Fig. 9: MeteoSwiss operational C-band radar at Pointe de la Plaine Morte (2920m asl), MXPol X-band mobile





radar (460m asl) and the ground instrument Multi Angle Snowflake Camera (MASC, Garrett et al. (2012)), placed at 2370m of altitude.

The second step of the verification is based on analyzing the influence of the proposed de-mixing method on the classification matching between two radars covering a certain common volume. Namely, we defined the vertical cross section sized 7 km in range and 2 km in height, being common for the Plaine Morte radar 227° and the MXPol radar 47° azimuthal RHI (see blue and orange segments in Fig. 9). Further on, we selected a period of 50 minutes, corresponding to 10 acquisitions by the Plaine Morte radar (PPIs, and therefore reconstructed RHIs) and 14 RHI acquisitions by MXPol radar (05h05 to 05h55 UTC, on 28th February 2017), where the stationarity in terms of proportions of dominant labeled hydrometeors could be assumed.

[Figure 9 about here.]

One of the acquisitions, illustrated in Figures 10a, shows a clear difference in the sampling volume size between two radars. Namely, although the range resolutions of two radars are indeed comparable (75m for the MXPol radar, and 83m for the PlaineMorte radar), the vertical cross section is significantly closer to the MXPol radar, which makes it having much better azimuthal resolution, as well. Entropy estimate in Figures 10b shows an overall significant rise in the entropy values for the Plaine Morte radar, with respect to the ones of the MXPol radar. Due to the expected increase in hydrometeor mixing with the increase of the radar sampling volume, this observation, supported by other analyzed acquisitions, confirms the crucial role of the parameterized entropy in detecting hydrometeor mixtures. It therefore confirms the plausibility of the proposed bin-based approach, intrinsically related to the concept and the definition of the entropy parameter.

The data being kept in polar coordinates, it was impossible to properly match the volumes. Thus, a more direct, quantitative way of proving the utility of the proposed de-mixing approach, comes down to the comparison of the proportions of detected classes in the entire cross section, before and after the de-mixing (Fig. 10c - 10e). The quantitative mismatching criterion is the simple measure of difference between the normalized proportions of hydrometeors seen by the MXPol radar ($PR_{MX}$) and by the Plaine Morte ($PR_{PM}$):

$$D = \frac{\sum_{i=1}^{4} PR_{MX}^i - PR_{PM}^i}{\sum_{i=1}^{4} |PR_{MX}^i|^0 \vee |PR_{PM}^i|^0}, \tag{10}$$

normalized by the number of classes detected either by one, or by both instruments (logical or operator, in the denominator). The value of $D$, averaged over all considered acquisitions is provided in the titles of Fig. 10c -10e.

In Fig. 10d the de-mixing is limited to the proportions of the first three dominant components, which tend to occupy the majority of the radar sampling volume, and are not probable to account for barely residual presence of a hydrometeor class. This comparison does not only show the improved matching, but also the presence of hydrometeors unidentified before the de-mixing. Extending the de-mixing to all eight components (VI being merged with CR) improves furthermore the matching, at the expense of the occurrence of residually present classes (WS in this case).

The share of the dominant class (Fig. 10f and 10g) would be the quantitative confirmation of the logical assertion made in the previous paragraph: in the case of a bigger radar sampling volume, with the proposed method we tend to infer more mixing.





We benefited from this configuration to check as well for the potential correlation between the entropy parameter/mixtures indicator $H$ and polarimetric parameters constituting the weather radar target vector (Eq. 1). The weak, but statistically significant linear correlation (at .05 significance level) of $-0.4$ for MXPol and $-0.17$ for the Plaine Morte radar is observed for the co-polar correlation ($\rho_{hv}$), which makes quite some sense given that $\rho_{hv}$ is often interpreted as a measure of heterogeneity.

[Figure 10 about here.]

### 3.1.3 Comparison with the ground based observations

The final stage of the verification is based on confronting the outcome of the de-mixing method with the classification of individual particles from the ground based instrument. The principle intuitively resembles the comparison with the 2DVD based classification (Grazioli et al., 2014) in the original classification paper (Besic et al., 2016). The classification of MASC images is described in Praz et al. (2017). Using the supervised machine learning approach, it distinguishes between small particles (SP), columnar crystals (CC), planar crystals (PC), combination of columnar and plate crystals (CPC), aggregates (AG) and graupel (GR), with the additional possibility of estimating degrees of riming and melting. In order to make the comparison possible, we formed the corresponding merged classes: CR (CC + PC + CPC), AG (AG), RP (GR), MS (any of the classes with the melting degree different from zero).

The setup is again based on considering the vertical cross section of the reconstructed RHI of the Plaine Morte radar, though this time in a slightly more restricted area of 4 km in the range direction, around the MASC (Figures 11a and 11b). In order to more severely satisfy the hypothesis of stationarity (in terms of proportions of dominant labeled hydrometeors), which allows us to properly average the de-mixing scores, we selected different periods across different events, summarized in Table 1. The quantitative matching criterion is identical to the $D$ defined in Eq. 10, with $Pr_{MX}$ being replaced by $Pr_{MASC}$.

[Figure 11 about here.]

[Table 1 about here.]

Though the quantitative evaluation of mismatching shows pretty good results of the classification itself, similarly to the second stage of the verification, we can still notice that in all six analyzed events, regardless of their duration, the applied de-mixing method improves the distribution matching (smaller $D$). The matching also systematically appears to be better if we consider all components, rather than the three dominant ones. The improvement in matching is also correlated to the measure of entropy, averaged over the event and the vertical cross section ($\mu(H)$). Namely, lower $\mu(H)$ means smaller probability of mixing, and therefore smaller space for the progress in terms of distribution matching.

In Fig. 11c - 11e we illustrate how the distributions actually compare across four merged classes. The presented event (no. 3 in Table 1), highlights the improved agreement in terms of concentration of aggregates (AG) and rimed ice particles (RP). It also shows the risk of overestimating residually present particles (MS in this case), in case of the extension to all eight de-mixing components (Fig. 11e).

In order to demonstrate the capability of the method to deal with the hydrometeor classes other than accentuated aggregated and rimed ice particles, we illustrate the comparison of distributions across four classes for the event 5 in Fig. 12. Namely,



it was the warmest day with a significant amount of precipitation during the campaign, and our hope was to perform some de-mixing around the melting layer. Unfortunately, though the MASC was effectively in the melting layer, the lowest radar beam was 400m-500m above the $0°$ isotherm, and MXPol data happen to be compromised by ground clutter in the direction of interest. Nevertheless, the de-mixing shows very good performances in estimating the proper percentage of the third class

involved - crystals (Fig. 12b and 12c).

[Figure 12 about here.]

## 4    Neighborhood-based analysis

The neighborhood-based analysis is founded on simultaneously considering an ensemble of pixels, rather than one pixel at a time as it was the case with the bin-based approach. This approach rises the issue of the incoherency in weather radar

measurements (Tso and Mather, 2009), the phenomenon which is conveniently neglected in the bin-based approach, even though it could potentially be considered relevant in the context of hydrometeor mixtures. Namely, given that the size of the hydrometeors populating the sampling volume is inferior to the size of the volume itself, the conventional radar measurements are affected by the random spatial interference of the scattered waves, the speckle effect. Averaging the parameters over several pulse responses i.e obtaining the estimates from time averages of auto and cross correlations of received echoes (Balakrishnan

and Zrnic, 1990), conventionally serves as a sort of speckle filter, which should deal with the potential incoherency. However, in the radar sampling volume populated by different hydrometeor types (characterized by significantly different geometrical shapes and fall velocities), some residual incoherency can be logically expected. Therefore, we decided to proceed with the following analysis, inspired by SAR incoherent polarimetric decompositions (Cloude and Pottier, 1996), and adapted to our specific polarimetric framework introduced in Section 2.1. It should be noted that the phase indicator, as external information,

is not included (phase indicator, the fifth element of our weather radar target vector (Eq. 1), meaning that from now on $\mathbf{k}$ is a fourth element vector).

### 4.1    PCA

Principal Component Analysis (PCA) is a statistical method which transforms the data represented in a space formed by correlated variables, to the space formed by orthogonal, linearly uncorrelated variables (Pearson, 1901). By assuming that our data

samples represent points in the space formed by four transformed and linearly scaled polarimetric variables ($[x_1\ x_2\ x_3\ x_4]^T$), the PCA comes down to the eigenvector decomposition of the sample estimated ($\langle \cdot \rangle$) covariance matrix of data samples. This allows us to describe the covariance matrix as a weighted sum of covariance matrices of the eigenvectors i.e. the principal components $\mathbf{k}_i$:

$$\langle \mathbf{k}\mathbf{k}^T \rangle = \left\langle [x_1\ x_2\ x_3\ x_4]^T\, [x_1\ x_2\ x_3\ x_4] \right\rangle =$$

$$= \lambda_1 \mathbf{k}_1^T \mathbf{k}_1 + \lambda_2 \mathbf{k}_2^T \mathbf{k}_2 + \lambda_3 \mathbf{k}_3^T \mathbf{k}_3 + \lambda_4 \mathbf{k}_4^T \mathbf{k}_4, \tag{11}$$



with the weights $\lambda_i$ being the corresponding eigenvalues. The obtained eigenvectors should represent scatterers, in our case types of hydrometeors, incoherently mixed in the considered data ($\mathbf{X}$). By projecting the original incoherent dataset values $\mathbf{X}$ onto the set of eigenvectors we obtain the dataset values in the new space - $\mathbf{Y}$. These new values could be taken for the measurements as they presumingly should be, if it were not for the incoherence in the measurements:

$$
\quad \mathbf{Y} = \begin{bmatrix} y_{11} & y_{12} & y_{13} & \cdots \\ y_{21} & y_{22} & y_{23} & \cdots \\ y_{31} & y_{32} & y_{33} & \cdots \\ y_{41} & y_{42} & y_{43} & \cdots \end{bmatrix} = [\mathbf{k_1}\ \mathbf{k_2}\ \mathbf{k_3}\ \mathbf{k_4}]^T \mathbf{X} =
$$

$$
= [\mathbf{k_1}\ \mathbf{k_2}\ \mathbf{k_3}\ \mathbf{k_4}]^T \begin{bmatrix} x_{11} & x_{12} & x_{13} & \cdots \\ x_{21} & x_{22} & x_{23} & \cdots \\ x_{31} & x_{32} & x_{33} & \cdots \\ x_{41} & x_{42} & x_{43} & \cdots \end{bmatrix}. \tag{12}
$$

Going backwards, by applying the inverse PCA transform, we can estimate the proportions of the originally measured samples contributing to each of the "pure" uncorrelated components:

$$
\mathbf{X}_i = \mathbf{k}_i [y_{i1}\ y_{i2}\ y_{i3} \cdots], \, i = 1, \cdots 4. \tag{13}
$$

**4.2  ICA**

Independent Component Analysis allows for a more rigorous separation of components with respect to PCA (Comon, 1994). Namely, in case of having non-Gaussian data the separation can be achieved at statistical moments higher than variance. That is to say, the principal components in this case are only uncorrelated, but not indeed independent. If we rely on the framework introduced in Eq. 12, and substitute the matrix of projected points $\mathbf{Y}$ with the matrix of independent sources $\mathbf{S}$, the vectors $\mathbf{k}_i$

take up the role of independent components:

$$
\mathbf{S} = \begin{bmatrix} s_{11} & s_{12} & s_{13} & \cdots \\ s_{21} & s_{22} & s_{23} & \cdots \\ s_{31} & s_{32} & s_{33} & \cdots \\ s_{41} & s_{42} & s_{43} & \cdots \end{bmatrix} = [\mathbf{k_1}\ \mathbf{k_2}\ \mathbf{k_3}\ \mathbf{k_4}]^T \mathbf{X} =
$$

$$
= [\mathbf{k_1}\ \mathbf{k_2}\ \mathbf{k_3}\ \mathbf{k_4}]^T \begin{bmatrix} x_{11} & x_{12} & x_{13} & \cdots \\ x_{21} & x_{22} & x_{23} & \cdots \\ x_{31} & x_{32} & x_{33} & \cdots \\ x_{41} & x_{42} & x_{43} & \cdots \end{bmatrix}. \tag{14}
$$

Their independence is reached by relying on the paradigm used in PCA (eigenvalue decomposition), but applied on tensorial structures, which are higher order generalizations of covariance matrices. Alternatively, it is done by means of an iterative

process aiming to increase the non-Gaussianity of the sources. In the latter case, adopted in this analysis, the hypothesis is that





due to the Central Limit Theorem, the increase in the non-Gaussianity of the sources will lead to the increase in their mutual independence (Hyvärinen and Oja, 2000). Therefore, an independent component is found as:

$$\mathbf{k}_i = \arg\max \mathbb{E}\left(f_{\text{ng}}\left([s_{i1}\ s_{i2}\ s_{i3}\ \cdots]\right)\right) =$$
$$= \arg\max \mathbb{E}\left(f_{\text{ng}}(\mathbf{k}_i^T \mathbf{X})\right), \tag{15}$$

5 with $f_{\text{ng}}(\cdot)$ being the measure of non-Gaussianity (most commonly kurtosis or negentropy). Basically, the vector $\mathbf{k}$ should be such that its product with the data samples $\mathbf{X}$ results in a variable $s$ with higher statistical moments (above the second) as pronounced as possible.

As it was the case with PCA (Eq. 13), we can go backwards, and estimate the contribution of the original samples to the each of "pure" independent components:

10 $$\mathbf{X}_i = \mathbf{k}_i\left[s_{i1}\ s_{i2}\ s_{i3}\cdots\right],\ i = 1,\cdots 4 \tag{16}$$

## 4.3 Evaluation

After elaborating in the previous section the de-mixing of aggregates, rimed ice particles and crystals through the bin-based approach, now we approach the same problem by trying to evaluate the potential lack of coherency. As previously stated, the polarimetric parameters characterizing a radar volume ($\mathbf{k}$) are already obtained through temporal averaging of subsequent radar 15 pulses, and therefore, the wider spatial context, determined by the values of entropy, is adopted here as a sort of polygon for the coherency study. Namely, the performance of PCA and ICA are studied on the example of MXPol and Plaine Morte datasets already used in Fig. 10, with a slightly more restricted surface, but a less restricted temporal stationarity constraint (a longer event).

[Figure 13 about here.]

20 In Fig. 13a we see the region of significant transitions between aggregates and rimed ice particles, seen simultaneously by the MXPol and the Plaine Morte radar, which is, due to the entropy estimation (thresholded at $H > 0.4$), suspected to be dominated by mixtures. The subsequent Figures 13b and 13c show the first (the closest centroid) and the second (the second closest centroid) dominant component as seen by the hydrometeor classification. Rather than analyzing separately each of the pixels and estimating the proportions of these components by applying the bin-based approach based on the assumption of 25 coherence, here we analyze the potential of the PCA and ICA techniques to detect the potential incoherence. That is to say, we try to infer the coherent component involved in the overall mixing process by considering the ensemble of the pixels, regardless their position in space.

We start by taking all the pixels, observed by the MXPol radar in one of the acquisitions during the extended event introduced in Section 3.1.2 (34 acquisitions between 04h00 and 06h00 UTC on 28th February 2017), and characterized by higher entropy 30 values ($H > 0.4$). By representing them in a space formed by our four transformed and stretched polarimetric variables, we obtain a not exactly informative cloud of points (black circles in Fig. 14). This cloud of points, which would be the matrix $\mathbf{X}$



introduced in Eq. 12 (each point is a matrix column), is characterized by relatively high entropy values, as this was the criterion for the selection of the region of interest. The following four figure columns represent the vectors $\mathbf{k}_i$, which are the axes of the new space i.e. the principal, uncorrelated components (in blue), or the independent components (in red).

[Figure 14 about here.]

As suggested in Subsection 4.1, the first uncorrelated components is supposed to represent a "pure" component, freed up of the effect of incoherence in the data acquisition. And indeed, this first component does cover for the majority of spatial variance (in Fig.14, the proportion of the first component is about 85 % for the PCA and 68 % for the ICA), indicating that we should not be too concerned by the residual backscattering incoherence. It is even more true if we recall that our sample cannot be considered as completely homogeneous, meaning that the portion of variance unexplained by the first uncorrelated component
does not exclusively refers to the incoherency. By repeating the analysis over the entire considered event, we confirm the that the proportion of the first component is important enough to discard a significant influence of residual backscattering incoherency (Fig. 15a), its median value being $86\%$, and its median Cloude and Pottier entropy value (Eq. 3, with $\lambda_i$ from Eq. 11 taking the role of $p_i$) being $\tilde{H}_{CP} = 0.38$. The low value of $H_{CP}$ emphasizes the non-uniform distribution among proportions of the four estimated principal components.

[Figure 15 about here.]

Unfortunately, by checking up the entropy estimate of the "pure" component in the space of the original centroids, we see that these vectors $\mathbf{k}_i$ do not correspond at all to the predefined centroids (not illustrated in Fig. 14 and 16). Hence, they cannot be considered at all as the "pure" components in the context of the hydrometeor classification. The conclusion is confirmed by checking the entropy distribution of the proportions of original data contributing to each of the components (Eq. 13). This
implies that PCA cannot be really used as a de-mixing tool, because the coherent backscattering proportion does not correspond to the backscattering of a "pure" hydrometeor type, but rather to the backscattering of the mixture.

Aside from uncorrelated components, in Fig. 14, we also show the results of employing the introduced FastICA method (with kurtosis value for the measure of non-Gaussianity), following its demonstrated benefits in the framework of the SAR decomposition theory (Besic et al., 2015; Pralon et al., 2016). The principal advantage of this tool with respect to PCA would
be the lack of the orthogonality constraint. Basically, each successive component is not required to be orthogonal to the previous one, but is the one which is genuinely independent, assuming the very plausible non-Gaussianity of the data. This effect, allowing for the subtle "splitting" of the first component, causes that the proportion of the first estimated component, event though it obviously corresponds to the first uncorrelated component, ends up being far inferior with respect to the one estimated by PCA (over the entire event, median proportion of $46.4\%$ and $\tilde{H}_{CP} = 0.86$, with $p_i$ being $||\mathbf{k}_1||_2^2$). This kind of
increased sensibility, which results in the subtle splitting of the first component, does not indicate higher incoherency than seen by PCA, but rather confirms the previously stated assumption that the incoherence is not the dominant cause for the proportion unexplained by the first correlated component.

[Figure 16 about here.]





The same analysis, applied on the Plaine Morte data (24 acquisitions between 04h00 and 06h00 UTC on 28th February 2017) is illustrated in Fig. 16. The quantitative parameters estimated over the entire event (PCA: median proportion of the 1st component 56%, $\tilde{H}_{CP} = 0.73$, ICA: $46.4\%$, $\tilde{H}_{CP} = 0.88$), show a drop in the first uncorrelated component proportion (Fig. 15). A drop which can be explained by the significantly increased volume size (which can be seen in Fig. 13), and a lower

number of pulses averaged in estimated the polarimetric parameters. Both factors logically make the hypothesis of coherency slightly weaker.

Aside from studying the potential effect of incoherency, this analysis is also useful to highlight the limit of the concept of discrete hydrometeor classification. Namely, this concept prevents us from exploiting the conventional tools in dealing with the residual incoherency, that could allow us to have an even more systematic and assumptions free insight into the mixed

radar sample volume, with respect to the one presented in this article as the bin based approach. A possible way forward would be to investigate the possibility of modeling the weather radar target vector. Following suggestions from the micro-physical modeling (Morrison and Milbrandt, 2015), or practices from the SAR remote sensing community (Touzi, 2007), one could develop a way to introduce some degrees of liberty in depicting hydrometeor classes, degrees of freedom which would reflect the physical properties of hydrometeor itself. That would not guarantee us to exploit all the statistical information from the

measurements, because inferring total independence is maybe not even possible (the proportions summing up to the unity could represent an obstacle to the concept of independence, as suggested by Nascimento and Dias (2005)). However, it could very probably allow us to account for the incoherency by exploiting the first two statistical moments (PCA).

## 5   Conclusions and future perspectives

In this paper, we address the issue of hydrometeor mixtures in polarimetric radar measurements by adapting the paradigm of

decomposition/unmixing widely elaborated in other remote sensing domains, to the field of weather radar remote sensing.

In the first part of the paper we propose a bin-based de-mixing approach, which is largely based on the hypothesis of coherent backscattering of hydrometeors inside the radar sampling volume. The proposed approach is built upon, the semi-supervised hydrometeor classification method which reduces the classification problem to the distances in the Euclidean space formed essentially by the polarimetric parameters (Besic et al., 2016), but could be adapted to any classification technique providing

a distance to the various hydrometeor types. Inspired by the SAR polarimetric decomposition on standard mechanisms and hyperspectral linear unmixing, the method estimates proportions of different hydrometeor classes in each radar sampling volume, without considering the wider spatial context. The performance of such an approach is analyzed in three stages, based on C-band and X-band radar data, together with a ground based Multi-Angle Snowflake Camera. The analysis, aside from demonstrating the potential of the method, also shows the improved matching between different radars, and most significantly,

the improved matching between the radar and the independent ground based instrument.

The second part of the paper is dedicated to the study of a potential influence of the residual incoherency in the backscattering of hydrometeors inside the radar sampling volume. The study is based on adapting the conventional statistical methods, as PCA and ICA, used to deal with the incoherency in the SAR remote sensing, to the specific framework of the weather radar



polarimetry. The performance analysis points out the limited influence of the residual incoherency in the regions of hydrometeor mixtures. This conclusion, implying that after all there is not a significant rise in incoherency in case of hydrometeor mixtures, on one side strengthens the proposed bin-based approach, and on the other side makes the tools as PCA and ICA less useful in the context of weather radar decomposition/de-mixing.

The overall message of this paper is to focus some attention of the weather radar community to the importance of the decomposition/de-mixing methods, which make it possible to look into the radar sampling volume. The present work remains exploratory, and many avenues still need to be explored, among which the potential benefit of a continuous hydrometeor classification approach. Finally, the proposed bin-based approach, allowing already plausible and fairly validated estimation of hydrometeor type at the sub-bin level, can be used to improve the quantitative estimation of precipitation using radar.

*Code and data availability.* Codes can be made available upon request to the authors. Datasets acquired by the MXPol X-band radar and the MASC can be made available up request to the authors. For data acquired by the operation C-band radar network contact the authors affiliated with MeteoSwiss..

*Author contributions.* NB and AB developed the concept of the paper, performed the analyses and interpreted the results. CP and JG particularly contributed to the aspect related to MASC acquisition and processing. JFV and JG notably contributed to the radar data acquisition

and processing part. UG and MG notably contributed to the conceptual and the interpretation segments. NB with contributions of all authors, prepared the manuscript.

*Competing interests.* The authors declare that they have no conflict of interest.

*Acknowledgements.* The authors would like to thank their colleagues at LTE and Radar, Satellite, Nowcasting teams for all their useful suggestions and their support in the data processing. Particularly, we would like to emphasize the help of Floortje Van den Heuvel and Peter

Speirs due to their indispensable role in the organization of the campaign of measurements in the canton of Valais.



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



## List of Figures





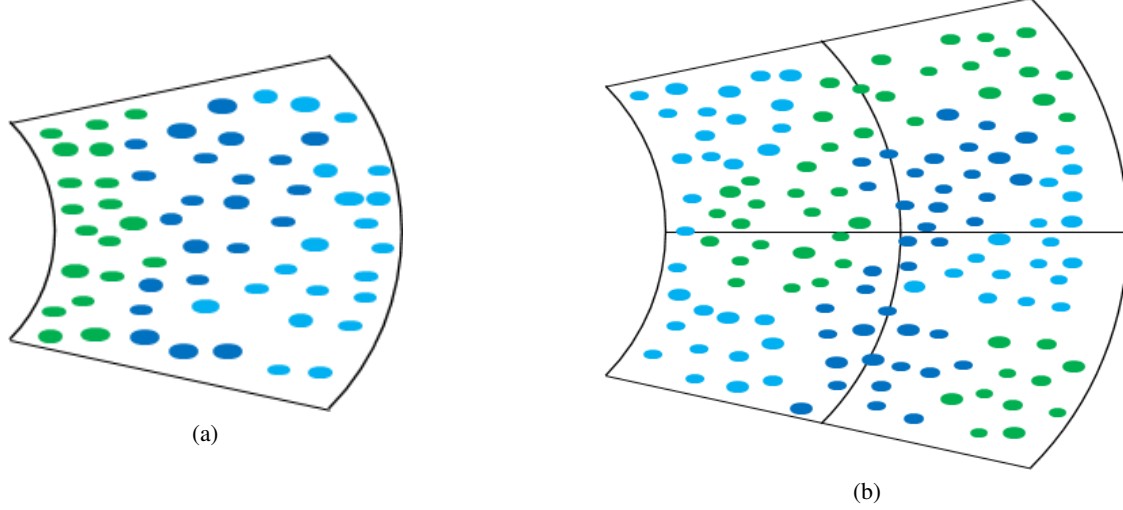

**Figure 1.** Simplified schematic representation of (a) bin-based de-mixing and (b) neighboring based analysis. In reality, the spatial organization of mixed hydrometeor types is presumed to be significantly more chaotic.





**Figure 2.** The multi-dimensional (four out of five dimensions) space: (a) target vectors representing centroids (larger points with different classes depicted by different colors) and observations (smaller black points), (b) observations with assigned labels. Abbreviations: CR - crystals, AG - aggregates, LR - light rain, RN - rain, RP - rimed ice particles, VI - vertically aligned ice, WS - wet snow, IH/HDG - ice hail/high density graupel, MH - melting hail.





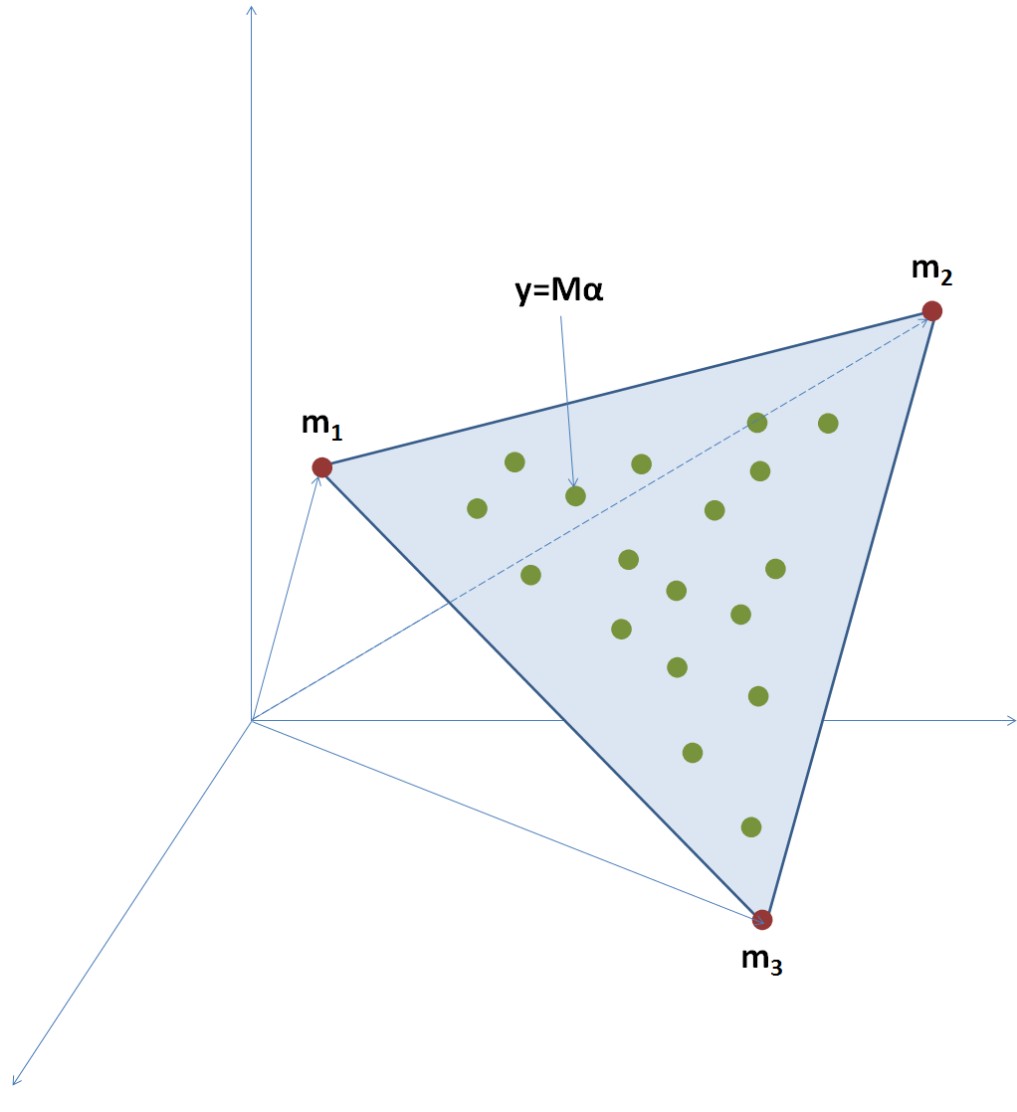

**Figure 3.** An example inspired by Bioucas-Dias et al. (2013) illustrating the problem of linear unmixing: blue simplex is defined by red $\mathbf{m}_i$ points depicting "pure" materials, whereas encompassed green points represent mixtures of these materials.



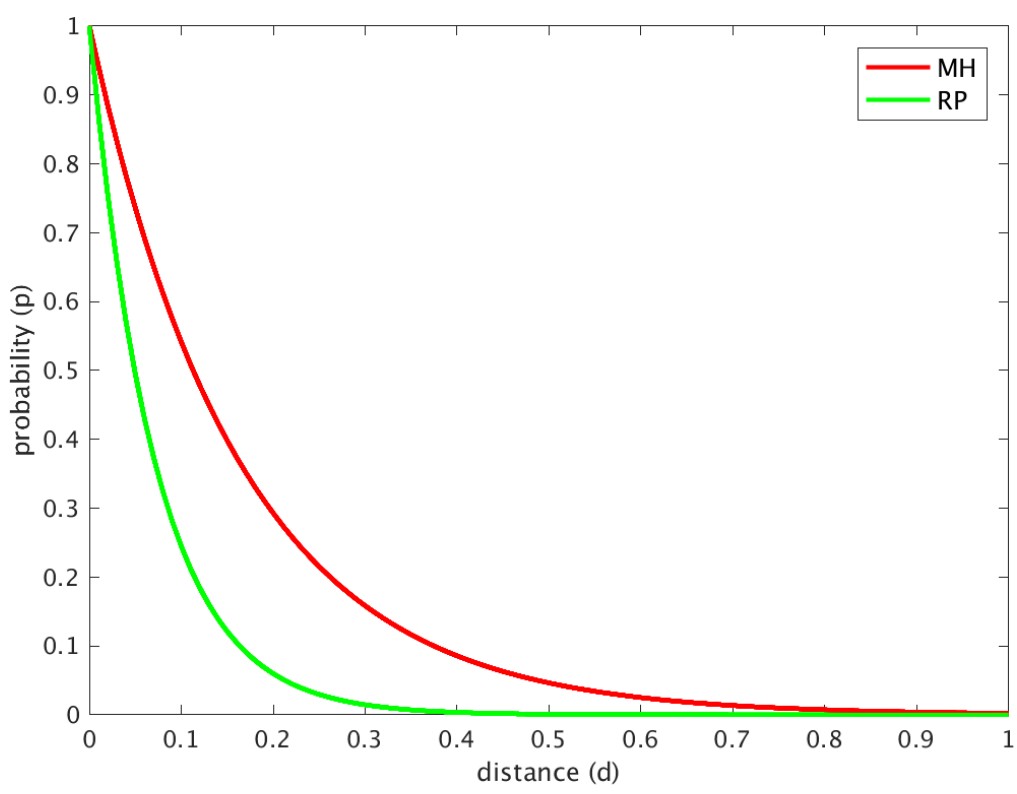

**Figure 4.** An example of the exponential transformation: the scaled distances to the probability ($d \rightarrow p$).





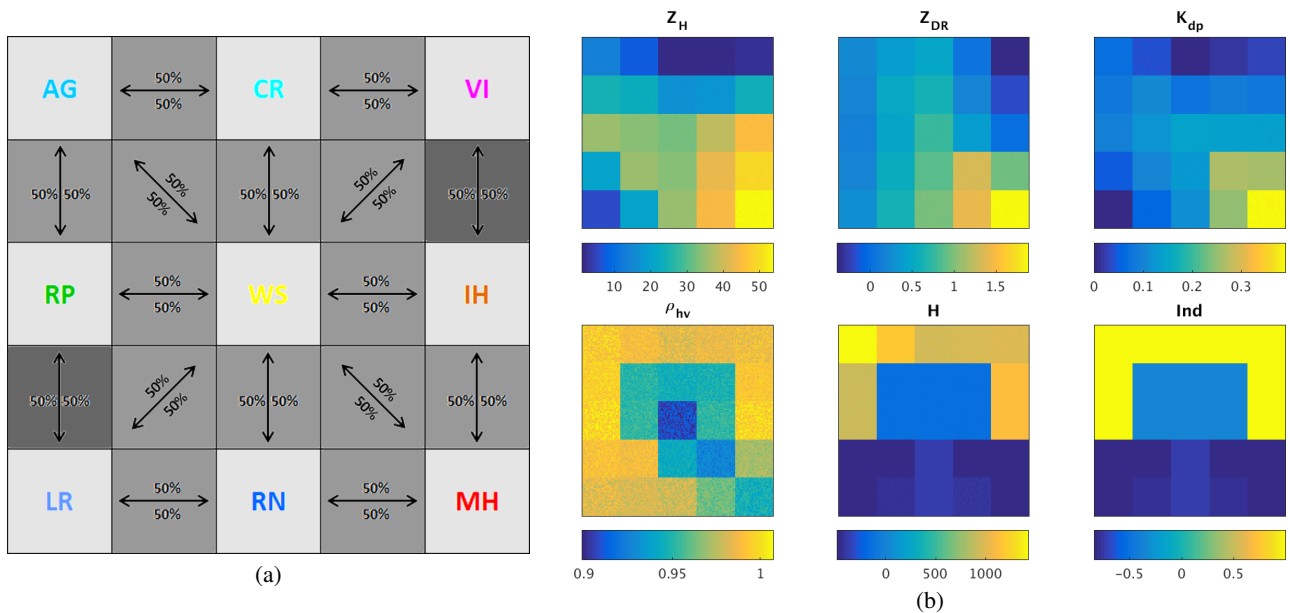

**Figure 5.** Quadratically organized synthetic dataset used in the entropy parametrization: (a) the plan with pure hydrometeors, mixtures, and not very plausible mixtures, (b) the value of elements of weather radar target vector (including the relative altitude with respect to the 0° isotherm - H).



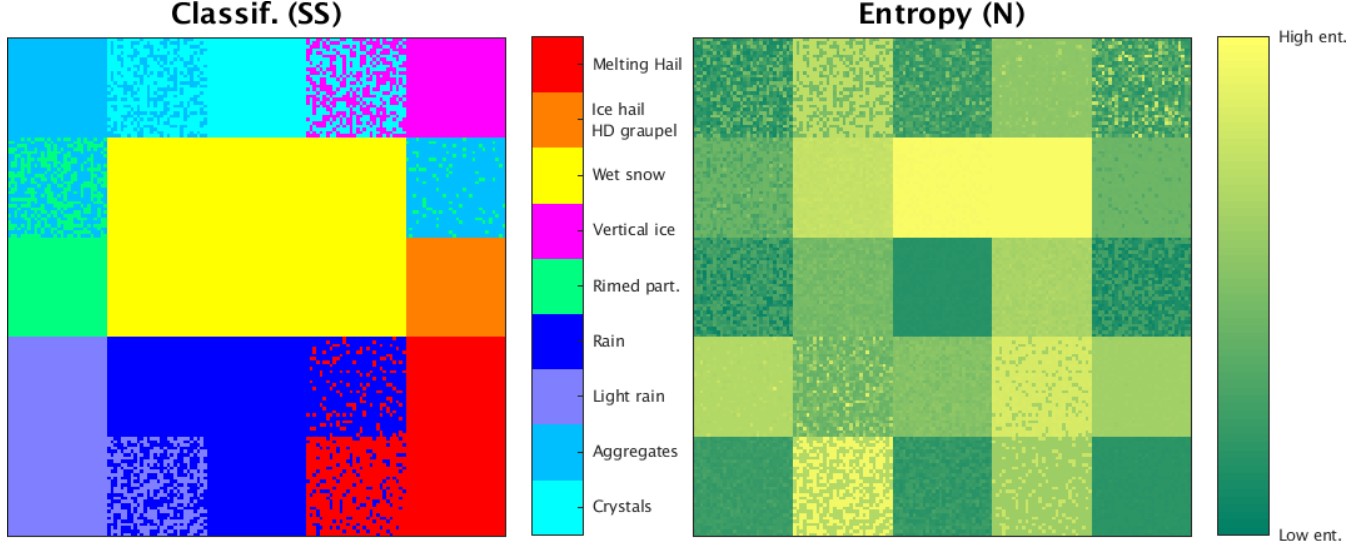

**Figure 6.** Classification and entropy applied on the synthetic dataset from Fig. 5. Low and high entropy values correspond to the limits of the range $[0, 1]$.





**Figure 7.** The multi-dimensional (four out of five dimensions) space, target vectors representing centroids (larger points with different classes depicted by different colors) and observations with the level of gray depicting: (a) Shannon version of the entropy from (Besic et al., 2016), (b) parametrized entropy.





**Figure 8.** Bin-based de-mixing applied on an example of MXPol dataset acquired: (1) during the APRES3 campaign at the Dumont-d'Urville base, Antarctica, on 28th January 2016, (2) during the HyMeX campaign in the region of Ardèche, France, on 24th September 2012; (a) classification followed by the entropy estimate, (b) proportion of eight hydrometeor classes in the each of radar sampling volumes.




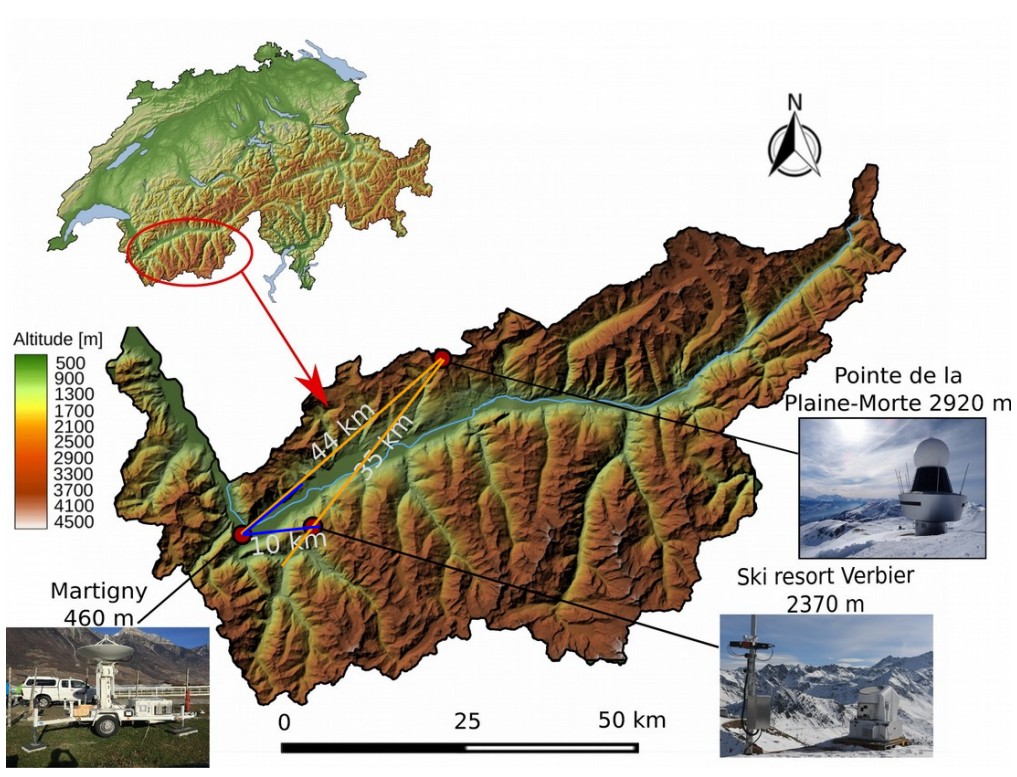

**Figure 9.** Configuration of instruments deployed during the Valais campaign.





**Figure 10.** Comparison example in terms of classification (a) and entropy (b) between Plaine Morte (1) and MXPol (2), followed by the quantitative matching analysis: (c) before de-mixing, (d) after de-mixing with only three dominant components, (e) after de-mixing with all components; and quantitative measure of mixing ratio: (f) with only three components, (g) with all components.



**Figure 11.** Comparison of classification applied on Plaine Morte data (a,b) with the MASC classification, before (c) and after de-mixing: (d) with the three dominant components, (e) with all components.



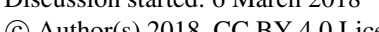



**Figure 12.** Comparison of classification applied on Plaine Morte data with the MASC classification, before (a) and after de-mixing: (b) with only three de-mixing components, (c) with all de-mixing components.



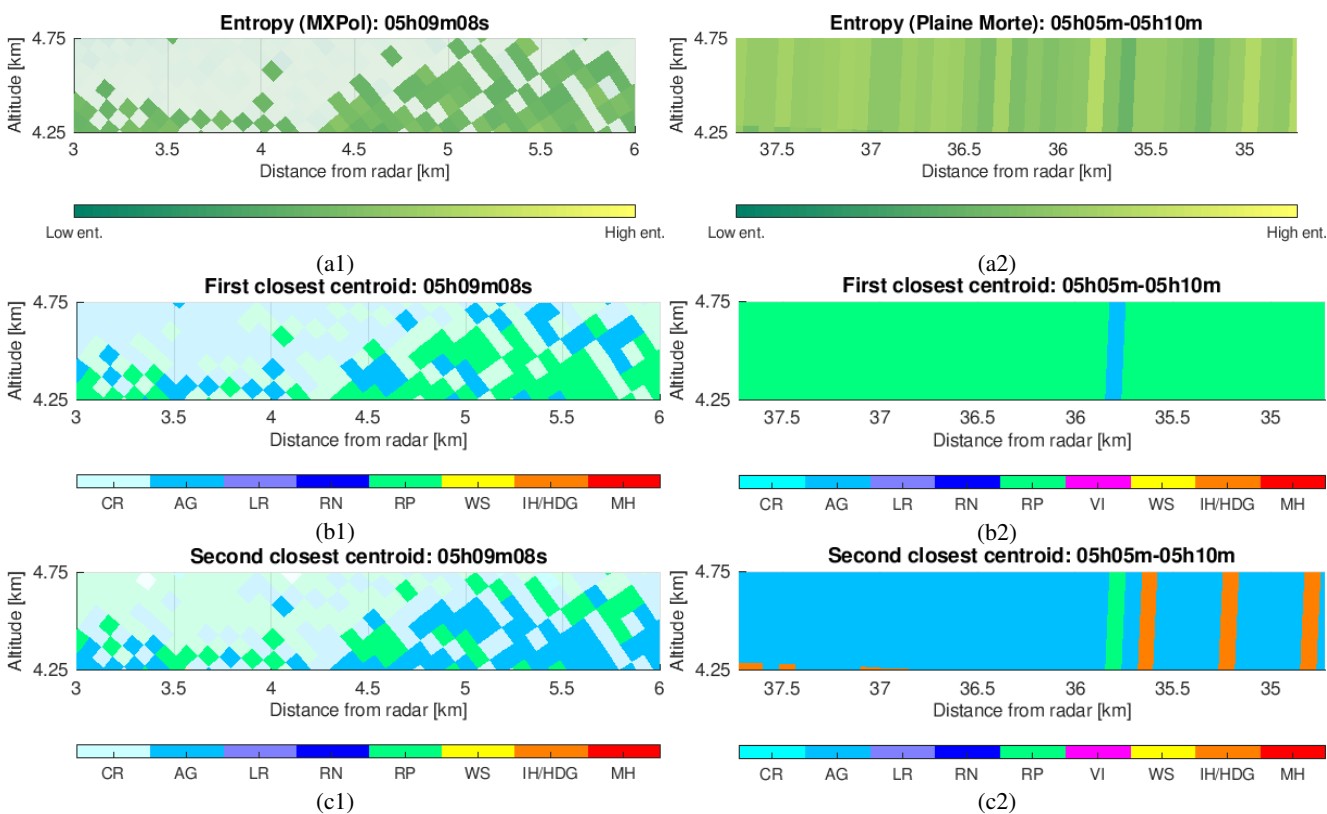

**Figure 13.** The example of (1) MXPol radar RHI and (2) Plaine Morte reconstructed RHI (from the dataset used in the Section 3), with the accentuated volumes with entropy $H > 0.4$: (a) the entropy of the region of interest, (b) the first closest centroid, (c) the second closest centroid .





**Figure 14.** The PCA and ICA applied on the $H > 0.4$ regions of an example of the RHI from the event considered in the inter-radar comparison in Section 3.1 (MXPol X-band data).




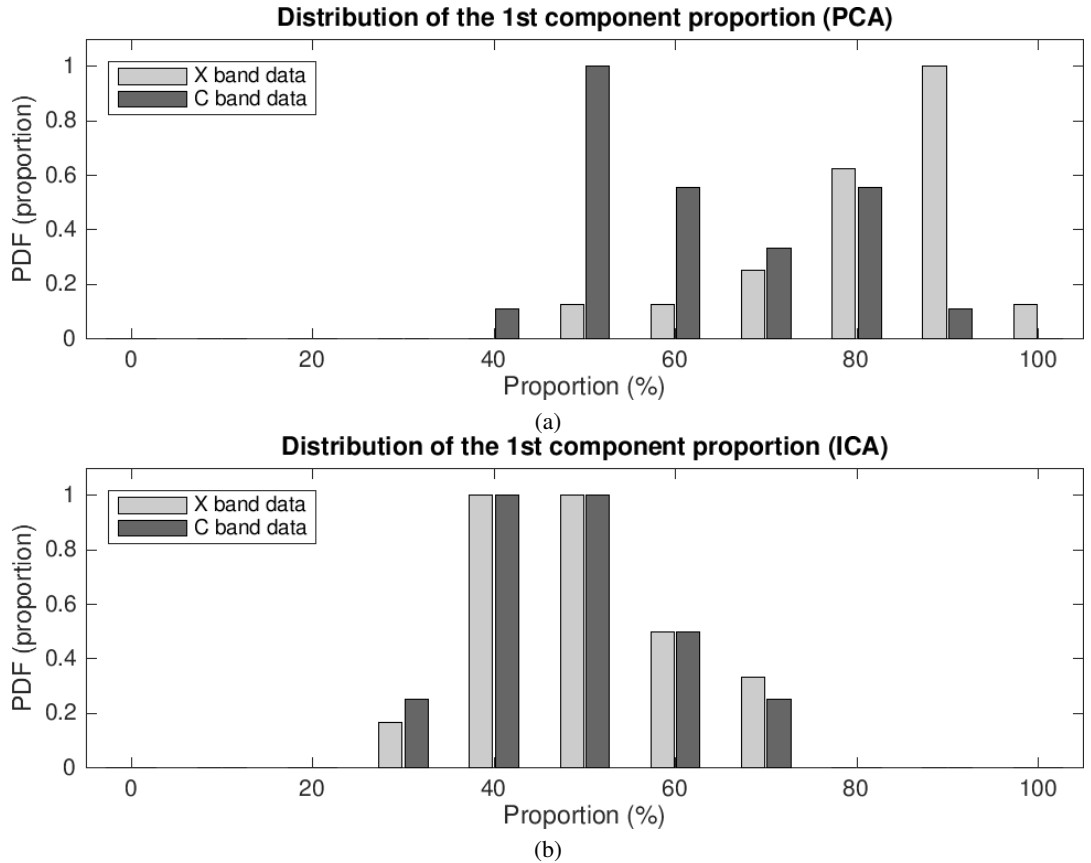

**Figure 15.** The distribution of the proportions of the most dominant component, calculated by generalizing the analysis illustrated in Fig. 13, 14 and 16, onto the entire (extended) event considered in the inter-radars comparison in Section 3.1: (a) PCA, (b) ICA.





**Figure 16.** The PCA and ICA applied on the $H > 0.4$ regions of an example of the RHI from the event considered in the inter-radar comparison in Section 3.1 (Plaine Morte C-band data).





**List of Tables**





**Table 1.** Quantitative scores ($D$) of the Plaine Morte vs MASC comparison before and after the bin-based demixing. Events 3 and 5 are illustrated respectively in Figures 11 and 12.

| No. | Event | $D$ bef. the de-mix. | $D$ aft. the de-mix. (3 comp.) | $D$ aft. the de-mix. (all comp.) | $\mu(H)$ |
|---|---|---|---|---|---|
| 1 | 12/01/17, 20h10-21h00 UTC | 0.1074 | 0.0977 | 0.0857 | 0.3303 |
| 2 | 05/02/17, 06h10-08h00 UTC | 0.2233 | 0.1372 | 0.1196 | 0.4445 |
| 3 | 06/03/17, 14h15-14h50 UTC | 0.0741 | 0.0376 | 0.0280 | 0.4091 |
| 4 | 06/03/17, 17h25-18h00 UTC | 0.3289 | 0.1954 | 0.1813 | 0.4510 |
| 5 | 18/03/17, 12h00-14h00 UTC | 0.1760 | 0.0859 | 0.0690 | 0.4429 |
| 6 | 18/03/17, 16h00-17h00 UTC | 0.3070 | 0.2092 | 0.1965 | 0.4290 |