# Peer review of "Unraveling hydrometeor mixtures in polarimetric radar measurements"

_Atmospheric Measurement Techniques, 2018_

## Referee Comment (RC1) · Anonymous Referee #1 · 18 Apr 2018

Title: Unraveling hydrometeor mixtures in polarimetric radar measurements

GENERAL COMMENT

This paper presents a statistically-based approach to reveal different hydrometeor types present within the same radar resolution volume (de-mixing). While general hydrometeor classification is devoted to the identification of the dominant particle type, the aim of this work is to use statistical techniques to further exploit the information provided by dual-polarization weather radar observations. The topic is of substantial interest for the radar community and the manuscript presents original contributions, building up on previous work by the same authors. My main concern is that the proposed method is heavily relying on statistics, without an in-depth consideration of the physics behind hydrometeor classification. I consider that some discussion about the

key physical factors should be introduced, in order to allow the reader to better understand and evaluate the relevance of the specific statistical techniques adopted in this context.

The manuscript in general well written, although the reading is sometimes made difficult, due to an exposition in my opinion unnecessarily complicated. I do not understand in particular the need for the SAR discussion (specifically, around eq. 4), since the proposed method is in the end substantially different (it does not consider the scattering matrix). This appears an unnecessary complication in the illustration of the method. The analogy with SAR may just be mentioned in the introduction. I consider that suppressing this discussion would be beneficial for the clarity of the exposition.

Literature: reference to other unsupervised or semi-supervised classification methods for weather radar could be included (e.g. Bechini and Chandrasekar, 2015; Weng et al., 2016), in addition to the authors previous paper, to provide a more general overview of the main topic. For a more physically-based de-mixing approach I also suggest mentioning the paper by Keat & Westbrook (2017), showing a physical de-mixing technique for the specific case of ice aggregates and pristine ice crystals.

The figure quality is good in general, with the exception of figure 8, which I recommend to split in two separate figures. The labels are way too small (especially for panel b1).

In the first part of the paper (bin-based de-mixing) a synthetic dataset is generated, "created by linearly mixing different pairs of hydrometeor classes in equal proportions". It is not clear how the polarimetric parameters are mixed, i.e. linearly mixing the polarimetric parameters is not the same as mixing in equal proportions hydrometeors, due to different scattering behaviors and varying sensitivity of the radar parameters to concentration, shape, density, orientation, etc. Please explain better. This is an example where the reader may be missing an adequate discussion about physical factors linking scattering, radar observations and classification.

For the second part of the paper (Section 4, the neighborhood-based analysis) I found

that the theoretical basis is not sufficiently clear. The two concepts of incoherency in backscattering (what about forward scattering, i.e. Kdp?) and correlation among radar moments are mixed without a clear definition and explanation of the relative background. The purpose of the method seems to use PCA and ICA to remove the residual correlation between radar moments, allowing to detect potential incoherency. Although it is well known that the co-polar correlation coefficient provides an indication about mixtures within the radar resolution volume, what should we expect about correlation between different radar moments? How do you justify the assumed relation with incoherency? I recommend to critically revise this part, providing more sounding arguments to support the presented approach. Alternatively, I suggest to consider dropping this part and focus on the first bin-based classification and de-mixing. In fact, this part alone could be a nice paper on its own.

MINOR COMMENTS / CORRECTIONS

- P2, L30: "hydrometeorly" sounds awkward

- P3, L26: Normalization of radar variables. What is the impact of the arbitrary choice of min-max values for the scaling? The radar parameters may span very different dynamic ranges in different events. Could this result in an event-dependent classification performance?

- P11, L21: "fourth element vector" -> "four element vector"

- P5, L9: define "POLSAR", later spelled as "PolSAR" (P6, L13)

- P10, L7: maybe "comparing" is more appropriate here, instead of "confronting"

- P14, L3: "The following four figure..". Provide figure number

REFERENCES

Bechini, R. and V. Chandrasekar, 2015: A Semisupervised Robust Hydrometeor Classification Method for Dual-Polarization Radar Applications. J. Atmos. Oceanic Tech-

nol., 32, 22–47, https://doi.org/10.1175/JTECH-D-14-00097.1

Keat, W. J., & Westbrook, C. D. (2017). Revealing layers of pristine oriented crystals embedded within deep ice clouds using differential reflectivity and the copolar correlation coefficient. Journal of Geophysical Research: Atmospheres, 122, 11,737–11,759. https://doi.org/10.1002/2017JD026754

Wen, G., A. Protat, P.T. May, X. Wang, and W. Moran, 2015: A Cluster-Based Method for Hydrometeor Classification Using Polarimetric Variables. Part I: Interpretation and Analysis. J. Atmos. Oceanic Technol., 32, 1320–1340, https://doi.org/10.1175/JTECH-D-13-00178.1

Wen, G., A. Protat, P.T. May, W. Moran, and M. Dixon, 2016: A Cluster-Based Method for Hydrometeor Classification Using Polarimetric Variables. Part II: Classification. J. Atmos. Oceanic Technol., 33, 45–60, https://doi.org/10.1175/JTECH-D-14-00084.1

---

## Referee Comment (RC2) · Anonymous Referee #2 · 19 Apr 2018

The paper discusses the issue of hydrometeor mixtures in the Dual-pol measurements using a statistical approach. The study is completely, if I may say that, "legitimate". Indeed, the hydrometeor classification looks for the dominant or the highest probable one and that it. The mixing in the volume radar is very important due to the geometry of measurement and the volume size and by consequence, it leads to a lot of approximation and errors especially in quantitative measurement. But for sure, the mixing remains completely normal and important as physical phenomena. It is really difficult to follow; sometimes there is a kind of displacement in conversation to other subjects which are related but does not give any boost to the paper. I was excited to see the title, but at the end, I am somehow disappointed. The statistical approach is good but it could be complemented by some changing in parametrization in the scan. What I

mean is, Authors can start for example with 2 degrees of resolution in the Azimuth angle then decrease it until the minimum possible value in the scan. Then the statistic will be very helpful. The same thing with the RHI from a large resolution to a smaller one. The first approach (the bin based one) looks better than the second one is much more difficult to understand it, I invite the author to add more fluidity and clearance to this part.

But in general, the paper follows the series of one of the main topic of the LHTE and Meteo Swiss. Elegant work, the validation could be better, with larger figures (I used 400% zooming to see something .. impossible with a printed version) and as I mentioned before, more fluidity is needed and maybe playing with the Azimuth and elevation resolutions.

---

## Referee Comment (RC3) · Anonymous Referee #3 · 20 Apr 2018

Summary:

The manuscript describes efforts to utilize a statistically based approach to hydrometeor classification and how it can be used to identify not only the primary hydrometeor type, but also a secondary classification of particle within the radar volume. Scientifically, this is significant in that understanding where mixtures of particle types occur in the real atmosphere is extremely important to a variety of research efforts. The ability of a hydrometeor classification to identify mixtures of particles would be a large improvement over the current method of singular particle types.

Major Comments: While it is obvious that the authors spent considerable time in this effort and in the formation of this manuscript, the paper is presented in a way that makes it difficult to read. The article is very dense, and contains a significant amount

of "jargon" and unnecessarily complicated explanations. This makes it difficult to follow through some explanations and generally distracts the reader from the point of the paper. This is primarily the case in the first two sections of the manuscript.

Part of the verification that is used in the manuscript is comparing observations between the statistical HCA retrieval and a ground based snowflake camera (MASC). While this is a good comparison, this is only valid at the surface and primarily only for solid precipitation particles. The radar volumes being examined extend well above the surface where no verification data exists. A discussion about the limitations of comparison with the MASC due comparing a point measurement at the surface with a volumetric measurement aloft needs to be made. Furthermore, there is no discussion about the potential change in the hydrometeor distribution from when it leaves the lowest radar volume and before it reaches the ground.

The figures need a substantial amount of work. Entropy is displayed on several figures (6, 7, 8, 10, 11, and 13) and is given a range of "low" to "high". This range is meaningless and uninformative. There needs to be a definable range for these values. Figure 8 is particularly busy and while the plots are of good quality, their size makes it impossible to examine. Break this figure up into at least two figures.

Minor Comments: Pg 2, line 6: This line is difficult to read, suggest rewrite as: "Hydrometeor classifications are a very popular topic in the weather radar community, particularly since dual polarization radar became a widely used technology (Bringi et al., 2007).

Pg 2, lines 9-12: This sentence is long and difficult to read, suggest breaking it up

Pg 3, line 29: Phv typically has a value range of 0-1, yet you list -50 - -5.23. Is this value range correct?

Pg 5, line 17 - "...prevents us from applying the..."

Pg 6, line 7 - "...one can notice the intuitive similarity of the problem." Don't make the

assumption that this is as easily viewable by the average reader. Explain why its similar

Pg 6, line 13 - PolSAR, previously it was POLSAR, be consistent with your acronym, also need to define what POLSAR is when it is first used.

Figure 5: Why is there text in an alternate color? The figure caption should all be a single color of text.

Figure 8: In addition to the point listed in the major comments, there need to be more tick marks on the altitude axes. There are a number of altitude axes in this figure that don't have any labels at all. This is also true regarding the proportion color bar which shows a scale of 0-100%. A intermediate value or two is needed here to determine the spacing of the scale. Suggest either 0,50,100% or 0,25,50,75,100%.

Figure 10: I understand the use of color here, but this should be stated in plain text rather than attempting to use color in the figure caption.

Figure 11: The labels of the axes are small and difficult to read. Please make these labels larger. This might be a complication of attempting to use 5 plots in a single figure, in which case I suggest breaking the figure up. Additionally, don't use text color in the figure caption, just state the color in words.

––––––––––––––––––––––––

---

## Referee Comment (RC4) · Anonymous Referee #4 · 25 Apr 2018

The paper "Unraveling hydrometeor mixtures in polarimetric radar measurements" introduces classification method which quantifies the dominant contributor to the polarimetric variables in the radar resolution volume and separates it from the secondary contributors. The authors refer to it as bin based de-mixing. The method is novel and merits publication. Nonetheless the paper can be improved and made friendlier to readers. I list in no particular order the more significant issues.

The first part of the paper can stand on its own and I don't see the reason for introducing the Principal Component Analysis, other than to show it is not possible to find linear combinations of the polarimetric variables mutually orthogonal. They state this fact as follows "This implies that PCA cannot be really used as a de-mixing tool, because the coherent backscattering proportion does not correspond to the backscattering of

a "pure" hydrometeor type, but rather to the backscattering of the mixture." They also write in the conclusion "there is not a significant rise in incoherency in case of hydrometeor mixtures, on one side strengthens the proposed bin-based approach, and on the other side makes the tools as PCA and ICA less useful in the context of weather radar decomposition/de-mixing." Therefore I suggest they eliminate the section on PCA and ICA. Concentrate on the first part and try to explain better how they do the de-mixing.

I have issues with the terminology "coherency in backscattering" and also with "potential incoherency in the backscattering from different hydrometeors". "Coherent backscattering" is a well-defined and accepted terminology. Check the internet and also look up the articles by Jamison and Kostinski: " Direct Observations of Coherent Backscatter of Radar Waves in Precipitation" and "Partially Coherent Backscatter in Radar Observations of Precipitation" (both in J. At. Science). Here is what they say "The results agree with the earlier conclusions in the previous work, namely that coherent scatter occurs in both rain and snow, that it is larger in snow than it is in rain, and that it can be significant at times." So "coherent backscattering" can occur, it is rare, and not considered in radar meteorology. To be significant the spacing of the drops (hydrometeors) should be comparable and or smaller than the wavelength. For example if the spacing is $\frac{1}{2}$ of wavelength there would be tremendous enhancement of the returned signal. This is not the case, and I am sure the authors do not mean that it is. They use "coherent" in a different context. So they must devote a paragraph or more to explaining what they mean. After reading the paper I still am not sure? It seems to me they mean the similar type of hydrometeors contribute to the polarimetric variables (like Zh, Zv etc) so that the powers of the backscattered signal from each scatterer add, and the cross product (mutual interactions) cause the variability which is reduce through signal processing.? It this is what they mean they should state so. If I am correct then they could use the term "dominant contribution" for the ones from two or more species, perhaps the dominant can be quantified by %, like if the total of discernible contributions is more than say 60% than these contributors are dominant, the rest is a residue.

For most part the writing style is but there are quite few awkward spots. Next are some examples and my preferences. Pleas avoid use of targets as these are not missiles, or planes but are hydrometeors or scatterers. Copolar is one word (see the IEEE standards). Check if "hydrometeorly" is an accepted word? The rhv is copolar correlation coefficient (not just correlation). Page 4: By equalizing – should be "by equating". More severely should be "more strictly". This approach "raises" not "rises" (your spell checkers did OK, but no spell checker is yet logical, so let's make one and get rich). Many of your sentences are way too long making these unnecessarily complicated. Example P 14 starting with "This effect, allowing . . .. Has about 58 words.

To be fair to Straka – and true, you should list, " Straka, J.M., 1996: Hydrometeor fields in a supercell storm as deduced from dual-polarization radar. Preprints, 18 Conference on severe Local Storms, San Francisco, AMS. p 551-554. In that paper the fuzzy logic is used, but the weighting functions are pulse like (had top) and are overlapping – which is the essence of fuzziness. And add an appropriate statement – of first use of the technique in pol classifications.

---

## Referee Comment (RC5) · Anonymous Referee #5 · 10 May 2018

Summary: Nikola Besic et al. This paper presents a technique for performing hydrometeor identification (HID) on polarimetric radar measurements. In particular, it demonstrates a technique to retrieve a mixture of hydrometeors within a single bin, rather than only returning the dominant hydrometeor type. Only being able to return the dominant hydrometeor type has been an Achilles heel of HID. I believe the paper does have a high level of significance. Additionally the figures, other than a few places mentioned below, are of high quality.

The writing is adequate, though some portions would benefit from some editing for clarity. The paper is split into two effective sections. The mathematics in the paper can be confusing to follow as notations and terminology seems to change as the paper progresses. This is caused in part by trying to pull in techniques from several different

[Figure]

fields. Overall while I believe the concept has merit, I have some issues with some of the base assumptions. My largest objection is that after identifying the dominant hydrometeor, that the second closest centroid represents the most likely mixture component. I think the radar variables forming correlated frame vectors rather than a true basis make this assumption require quite a bit of justification.

The comparison of the data with the MASC is a good first step down this path, but as it reports fewer number of hydrometeors (3 ice classified in this case) than the classifier, it makes it hard to know for sure if there is any type of interference that would cause misclassification. I'm not convinced the authors have truly demonstrated that their secondary mode classification works. I would like to see potentially another dataset brought in to show some generality to the method.

While my review is somewhat critical, I do think the technique has a fair bit of merit and I would like to see the authors continue it through. In particular, I would love to see a way to address the issue of proportions of secondary components. The authors do equal mixtures for the development of the thresholding of the probability scores, but I believe one could do some simulation to give actual error bars on the retrieval by varying the components of the mixture at different amounts and then performing the retrieval. While real observations always need to be used as a final step, I would have liked to see more simulation data used to characterize the behavior of this retrieval.

Major Comments: From the abstract: "Intrinsically related to the concept of entropy, introduced in the context of the radar hydrometeor classification in Besic et al. (2016), the proposed method, based on the hypothesis of coherency in backscattering, estimates the proportions of different 10 hydrometeor types in a given radar sampling volume " As far as I can tell, entropy is not actually used in any of the proposed method, but rather as a followup parameter to further support the conclusions of mixed hydrometeor types in the bins.

This paper starts with an analogy to the Cloud and Pottier work on classifying polarimetric SAR, and then attemps to expand this to weather radar using reflectivity, differential reflectivity, Kdp and Rhohv as well as a phase indicator variable. I would like to echo the other commenters that the coherent and incoherent terminology should be explained in text due to its similarity to other common uses of that term in radar.

The exposition on page 5 related to POLSAR, while interesting, obscures the discussion of the technique as it applies to weather radar. While entropy is used, the Cloude and Pottier work uses much more than just entropy for their classification, namely adding anisotropy and alpha angle which provide useful information for the POLSAR classification. Those do not necessarily apply here (or at least are not explored in this paper). The switching between different notation and vocabulary can make the work difficult to follow.

I think the approach and workflow could be made much more clear. The classifier listed in this paper is essentially comparing the distance of a measured observation vector to each of the examplars and the minimum distance is assigned to that bin (equation 1). My biggest problem with the work is because the vectors of the space you are embedding into are not orthogonal, you can add 2 hydrometeor types to find a third type. So while I buy that the classifier predicts the dominant type, assuming that the second most likely hydrometeor is present in the mixture feels like a big leap and is not supported in the development of this work. Also what does proportion mean here? I understand it is somewhat of a mathematical construct here, but do the authors feel it is most akin to weight, number, or possibly relative amplitude of the signal returns?

On page 14 the statement : "As suggested in Subsection 4.1, the first uncorrelated components is supposed to represent a "pure" component, freed up of the effect of incoherence in the data acquisition "

Page 6.5-20: There is a lot of switching of terminology here between standard mechanism, end member, centroid, etc.

Page 6.11-12: "Nevertheless, basing the estimates of proportions on the distance in the

space populated by the standard mechanisms and measurements is a reasonable way to sum up the standard mechanisms." I don't feel like this has actually been justified and is a key premise of this work.

"Now that we have identified our centroids as standard mechanisms (analogy with Pol-SAR) and have found a mode for their addition via the distances in the Euclidean space of the classification " Again, I feel like this is overstated. The reduction in equation 5 and 6 of this to a linear mixing problem does not feel justified. Perhaps this is because $m_i$ is not actually defined here. I assume this is the observations matrix (k) associated with that pure material, but maybe I misread this?

"Boxes of hydrometeor mixtures contain different versions 5 of mostly plausible mixtures obtained by linearly combining in equal shares polarimetric parameters of two hydrometeor types involved, following assumptions of coherence and linearity " I would like to see more text as to how this mixing was accomplished. Figure 5 implies you only used equal mixtures of the two hydrometeors (50/50). Why limit it only to equal mixtures? I assume it has to do with finding the threshold value between two centroids that exists at exactly the 50% crossover point, but this should be made more clear.

Figure 1: I don't understand what the authors are attempting to demonstrate with this figure.

Figure 2: The definition of hydrometeor types in this paper is only in this figure caption. I would add a table or descriptive text with this information.

Equation 7: This $c_i$ is essentially the same as $m_i$ above correct? Maybe some clarifying statements here, or normalizing the notation would help. In general many of the equations seem to use one set of notation to introduce an equation (I assume the notation from that field), then apply it using a entirely different set of notation. You should unify the different representations.

Page 10: In the comparisons with the MASC, where are the centroids for each HID

class drawn from? In particular, are the centroids trained on this dataset, or on another dataset? I'm wondering how much overfitting may be involved in this process that would limit it's generalizability.

Minor Comments: Page 3.25 – leptokurticity, while a valid term, will cause many readers to have to stop reading and go look this term up as it is not commonly used in this field. Perhaps add a subclause to describe the term.

Figure 5: It took a few readings to understand that the greyscale level of the boxes and the text in (a) are related. Maybe make this more clear.

---

## Author Comment (AC1) · 11 Jul 2018

Please see our responses in the enclosed supplement.

Please also note the supplement to this comment:
https://www.atmos-meas-tech-discuss.net/amt-2018-58/amt-2018-58-AC1-supplement.pdf

―――――――――――――

---

## Author Comment (AC2) · 11 Jul 2018

ATMOSPHERIC MEASUREMENT TECHNIQUES

**Unraveling hydrometeor mixtures in polarimetric radar measurements**

N. Besic, J. Gehring, C. Praz, J. Figueras i Ventura, J. Grazioli, M. Gabella, U. Germann, and A. Berne

Dear reviewers, Dear Editor,

Thank you very much for all your efforts to improve the quality of the presented work. As you will hopefully notice in the following pages, we did our best to carefully reply to all of your questions and concerns, as well as to incorporate the resulting modifications into the manuscript.

The questions (**M** - major, **m** - minor, **S** - specific) and comments (**C**) of reviewers are reported in italic font. The modifications of the manuscript are highlighted in bold font.

The Authors

**1 Anonymous reviewer #1**

**C1:** *This paper presents a statistically-based approach to reveal different hydrometeor types present within the same radar resolution volume (de-mixing). While general hydrometeor classification is devoted to the identification of the dominant particle type, the aim of this work is to use statistical techniques to further exploit the information provided by dual-polarization weather radar observations. The topic is of substantial interest for the radar community and the manuscript presents original contributions, building up on previous work by the same authors. My main concern is that the proposed method is heavily relying on statistics, without an in-depth consideration of the physics behind hydrometeor classification. I consider that some discussion about the key physical factors should be introduced, in order to allow the reader to better understand and evaluate the relevance of the specific statistical techniques adopted in this context.*

The method, as presented in the paper, is primarily built upon the semi-supervised classification which itself simultaneously considers physics and data particularities (as elaborated in [Besic et al., 2016]). Therefore, it is the configuration of centroids which implicitly contains physics and guides the de-mixing procedure. Of course, in a more general approach, centroids do not have to be obtained by the quoted semi-supervised classification, but they however should always reflect presumed physical properties of different hydrometeors.

**C2:** *The manuscript in general well written, although the reading is sometimes made difficult, due to an exposition in my opinion unnecessarily complicated. I do not understand in particular the need for the SAR discussion (specifically, around eq. 4), since the proposed method is in the end substantially different (it does not consider the scattering matrix). This appears an unnecessary complication in the illustration of the method. The analogy with SAR may just be mentioned in the introduction. I consider that suppressing this discussion would be beneficial for the clarity of the exposition.*

Indeed, we may have exaggerated a bit with the part dealing with the PolSAR decompositions, though our intention was rather good: to provide a context and converge toward the de-mixing framework.

The revised version does not contain anymore the equation with the decomposition of the scattering matrix (Eq. 4 in the original manuscript).

**C3:** *Literature: reference to other unsupervised or semi-supervised classification methods for weather radar could be included (e.g. Bechini and Chandrasekar, 2015; Weng et al., 2016), in addition to the authors previous paper, to provide a more general overview of the main topic. For a more physically-based de-mixing approach I also suggest mentioning the paper by Keat and Westbrook (2017), showing a physical de-mixing technique for the specific case of ice aggregates and pristine ice crystals.*

The suggested papers are now properly cited.

**Page 2: In an effort to find a combination of these two conceptually opposed ideas, a series of semi-supervised methods was proposed [Bechini and Chandrasekar, 2015, Wen et al., 2015, Wen et al., 2016, Besic et al., 2016]. The one we rely upon in this work, presented by [Besic et al., 2016], clusters the data by including the hypotheses about the microstructure and the microphysics as a constraint.**

**Page 2: ...or for ice aggregates and pristine ice crystals ([Keat and Westbrook, 2017]).**

**C4:** *The figure quality is good in general, with the exception of figure 8, which I recommend to split in two separate figures. The labels are way too small (especially for panel b1).*

A very justified remark. The figure is now split into two figures (Fig. 8 and 9 in the revised manuscript).

**C5:** *In the first part of the paper (bin-based de-mixing) a synthetic dataset is generated, "created by linearly mixing different pairs of hydrometeor classes in equal proportions". It is not clear how the polarimetric parameters are mixed, i.e. linearly mixing the polarimetric parameters is not the same as mixing in equal proportions hydrometeors, due to different scattering behaviors and varying sensitivity of the radar parameters to concentration, shape, density, orientation, etc. Please explain better. This is an example where the reader may be missing an adequate discussion about physical factors linking scattering, radar observations and classification.*

Indeed, here we were rather linearly mixing the polarimetric parameters, which is now stated more explicitly. In the revised version, we also tried to elaborate the biases i.e. the systematic errors present in case of having unequal proportions of mixing hydrometeors.

**Page 7: Linear combination here stands for the arithmetic mean (equal proportions) in each dimension of the five-dimensional classification space.**

We principally agree that a discussion concerning physical factors linking scattering, radar observations and classification would not be redundant in an article dealing with the hydrometeor de-mixing/continious hydrometeor classification. However, this issue is pretty thoroughly addressed in the first paper [Besic et al., 2016] as well as in the other provided references dealing with the hydrometeor classification. In what concerns the de-mixing itself... in the revised version we tried to make it even more explicit that, a bit like in the case of hyperspectral remote sensing, the proposed approach has a fairly empirical character, the transformation providing proportions being parametrized using a synthetic dataset. Namely, the lack of the basis orthogonality prevents us from proposing a more solid, rational and analytical way of de-mixing (like the evoked coherent PolSAR decomposition), which could be a bit more "transparent" in terms of physics involved.

**Page 7: It would be indeed even more precise to combine the mixing components at the level of the electromagnetic scattering, before the integration leading to the employed polarimetric parameters. However, not event this would help us overcome the unavoidable lack of methodological "transparency" in terms of physics, due to the absence of the orthogonal de-mixing basis. The proposed method is therefore rather defined in a more empirical fashion, in the data processing plane, with the physical trustworthiness being verified experimentally, mostly using the independent measurements.**

**C6:** *For the second part of the paper (Section 4, the neighborhood-based analysis) I found that the theoretical basis is not sufficiently clear. The two concepts of incoherency in backscattering (what about forward scattering, i.e. Kdp?) and correlation among radar moments are mixed without a clear definition and explanation of the relative background. The purpose of the method seems to use PCA and ICA to remove the residual correlation between radar moments, allowing to detect potential incoherency. Although it is well known that the co-polar correlation coefficient provides an indication about mixtures within the radar resolution volume, what should we expect about correlation between different radar moments? How do you justify the assumed relation with incoherency? I recommend to critically revise this part, providing more sounding arguments to support the presented approach. Alternatively, I suggest to consider dropping this part and focus on the first bin-based classification and de-mixing. In fact, this part alone could be a nice paper on its own.*

The theoretical assumptions regarding the neighborhood based approach are now slightly more elaborated and hopefully more clear. As the reviewer correctly assumed, what we call here the spatial incoherency in scattering (backscattering or forward scattering) is indeed related to (diagnosed by) the correlation between radar moments. The whole idea is that by considering the spatial ensemble of integrated radar responses over radar sampling volumes, we can judge on the level of the random interference of the individual hydrometeor responses inside a radar sampling volume, surviving the averaging over several consecutive pulses. The co-polar correlation coefficient itself does not entirely substitute this kind of "diagnostics".

**Pages 12 and 13:** The neighborhood-based analysis is founded on simultaneously considering an ensemble of pixels, rather than one pixel at a time as it was the case with the bin-based approach. This approach rises the potential issue of the spatial incoherency in weather radar measurements [Tso and Mather, 2009], the phenomenon which is conveniently neglected in the previously presented bin-based approach, even though it could potentially be considered relevant in the context of hydrometeor mixtures. Namely, given that the size of the hydrometeors populating the sampling volume is inferior to the size of the volume itself, the conventional radar measurements are affected by the random spatial interference of the scattered waves, (otherwise known as the speckle effect), causing the incoherency in the measurements. It is important to state that the precipitation has been also characterized as a partly coherent scatterer [Jameson and Kostinski, 2010b, Jameson and Kostinski, 2010a], even though the former is more commonly associated to the clutter response [Zhang, 2016]. Averaging the parameters over several pulse responses i.e obtaining the estimates from time averages of auto and cross correlations of received echoes [Sauvageau, 1982, Balakrishnan and Zrnic, 1990], conventionally serves as a sort of speckle filter, which should deal with the incoherency by canceling the random interferences. This technique is presumed to be efficient in the case of originally incoherent measurements, whereas in the case of non-random interferences (coherent scatterer), its usefulness remains questionable.

In the radar sampling volume populated by different hydrometeor types (characterized by significantly different shapes and fall velocities), it is logical to suspect that some residual interferences could "survive" the conventional averaging over several pulses. Embracing the hypothesis of originally incoherent measurements, one could consider this to be the residual incoherency, though this could be equally be an intrinsically coherent backscattering described by [Jameson and Kostinski, 2010b, Jameson and Kostinski, 2010a]. However, in the context of de-mixing, only the residual incoherency is really potentially compromising the proposed bin-based approach, which is fundamentally based on the first order statistics - the vectors representing centroids. That is to say, the presence of incoherency in backscattering would require relying on the at least second order statistics in evaluating a mixture.

This being said, we decided to proceed with the following analysis, inspired by SAR incoherent polarimetric decompositions [Cloude and Pottier, 1996], and adapted to our specific polarimetric framework introduced in Section 2.1. The actual idea behind is that by considering the spatial ensemble of integrated radar responses over radar sampling volumes, we can assess the level of the "surviving" random interference of individual hydrometeor responses inside a radar sampling volume. Therefore, the following analysis can be considered as a way of diagnosing, by means of assessing the spatial consistency, the presence of the hereby described residual spatial incoherency. It is important to keep in mind that this technique most probably cannot distinguish the very small scale coherency in backscattering from the residual incoherency.

**m1:** *P2, L30: "hydrometeorly" sounds awkward.*

Indeed, it did sound awkward. It is now re-phrased as:

**Page 2:** ... in the case of radar sampling volumes with mixed hydrometeors...

**m2:** *P3, L26: Normalization of radar variables. What is the impact of the arbitrary choice of min-max values for the scaling? The radar parameters may span very different dynamic ranges in different events. Could this result in an event-dependent classification performance?*

The choice of min-max values employed in the scaling is not really arbitrary, but radar based on analyzing the distributions of polarimetric parameters acquired over 8 different events by radar, as it is elaborated in [Besic et al., 2016].

**m3:** *P11, L21: "fourth element vector" -¿ "four element vector".* ✓

**m4:** *P5, L9: define "POLSAR", later spelled as "PolSAR" (P6, L13).* ✓

**m5:** *P10, L7: maybe "comparing" is more appropriate here, instead of "confronting".* ✓

**m6:** *P14, L3: "The following four figure..". Provide figure number.* ✓

**2 Anonymous reviewer #2**

**C1:** *The paper discusses the issue of hydrometeor mixtures in the Dual-pol measurements using a statistical approach. The study is completely, if I may say that, "legitimate". Indeed, the hydrometeor classification looks for the dominant or the highest probable one and that it. The mixing in the volume radar is very important due to the geometry of measurement and the volume size and by consequence, it leads to a lot of approximation and errors especially in quantitative measurement. But for sure, the mixing remains completely normal and important as physical phenomena. It is really difficult to follow; sometimes there is a kind of displacement in conversation to other subjects which are related but does not give any boost to the paper. I was excited to see the title, but at the end, I am somehow disappointed. The statistical approach is good but it could be complemented by some changing in parametrization in the scan. What I mean is, Authors can start for example with 2 degrees of resolution in the Azimuth angle then decrease it until the minimum possible value in the scan. Then the statistic will be very helpful. The same thing with the RHI from a large resolution to a smaller one. The first approach (the bin based one) looks better than the second one is much more difficult to understand it, I invite the author to add more fluidity and clearance to this part.*

Following the suggestions of the reviewer we tried to reduce the content related to the other domains (PolSAR and hyperspectral remote sensing), in order to make the discourse more monotonous and easier to follow.

Unfortunately, at the moment we do not have any possibility to manipulate the scanning strategy. Namely, we do not have any ongoing campaign at the moment, and hence all the presented analysis are performed on the previously acquired data. Introducing the "post-scan" integration would actually be intuitively similar to the experiments with synthetic datasets we already use in the parametrization, particularly the one illustrated in Fig. 8 in the revised manuscript (Fig. 2 in this document).

Indeed, the second part (neighborhood based approach) introduced some confusion and we therefore appropriately elaborated it in the revised version. Please refer to #1.C6 and #4.C2 for more details.

**Page 9: Not having the possibility to retroactively manipulate the scanning strategy, we decided to base the second step of the verification on analyzing the influence of the proposed de-mixing method on the classification matching between two radars covering a certain common volume.**

**C2:** *But in general, the paper follows the series of one of the main topic of the LHTE and Meteo Swiss. Elegant work, the validation could be better, with larger figures (I used 400% zooming to see something .. impossible with a printed version) and as I mentioned before, more fluidity is needed and maybe playing with the Azimuth and elevation resolutions.*

Some of the figures, estimated as too charged with the information, are now reorganized to facilitate their interpretation (e.g. Fig. 8 in the original manuscript).

**3 Anonymous reviewer #3**

The manuscript describes efforts to utilize a statistically based approach to hydrometeor classification and how it can be used to identify not only the primary hydrometeor type, but also a secondary classification of particle within the radar volume. Scientifically, this is significant in that understanding where mixtures of particle types occur in the real atmosphere is extremely important to a variety of research efforts. The ability of a hydrometeor classification to identify mixtures of particles would be a large improvement over the current method of singular particle types.

**M1:** *While it is obvious that the authors spent considerable time in this effort and in the formation of this manuscript, the paper is presented in a way that makes it difficult to read. The article is very dense, and contains a significant amount of "jargon" and unnecessarily complicated explanations. This makes it difficult to follow through some explanations and generally distracts the reader from the point of the paper. This is primarily the case in the first two sections of the manuscript.*

We acknowledge that there is indeed some perhaps unnecessarily applied "jargon" and we tried to rectify this in the revised version... particularly in the first two sections.

**M2:** *Part of the verification that is used in the manuscript is comparing observations between the statistical HCA retrieval and a ground based snowflake camera (MASC). While this is a good comparison, this is only valid at the surface and primarily only for solid precipitation particles. The radar volumes being examined extend well above the surface where no verification data exists. A discussion about the limitations of comparison with the MASC due comparing a point measurement at the surface with a volumetric measurement aloft needs to be made. Furthermore, there is no discussion about the potential change in the hydrometeor distribution from when it leaves the lowest radar volume and before it reaches the ground.*

A fair remark, indeed. We reinforced the part dealing with the MASC based verification by explicitly stating the limitation of the instrument and commenting the changes potentially occurring between the radar lowest elevation and the ground level.

**Page 10: Evidently, neither the MASC as the instrument nor the classification applied on its measurements, can be considered as an ideal reference. The major limitations which ought to be considered when analyzing the results presented in this section are: the possible occurrence of blowing snow, the limited sampling volume of the instrument, the quality of recorded images, as well as the inevitable classification random errors.**

**Page 11: The unavoidable difference in height between the MASC and the lowest (non clutter contaminated) considered radar sampling volume could hypothetically compromise the comparison given the microphysical processes which can occur in this non-observed region. However, given the relative nature of the comparison (before and after de-mixing), as well as the employed spatial and temporal averaging, this effect is fairly limited.**

**M3:** *The figures need a substantial amount of work. Entropy is displayed on several figures (6, 7, 8, 10, 11, and 13) and is given a range of "low" to "high". This range is meaningless and uninformative. There needs to be a definable range for these values. Figure 8 is particularly busy and while the plots are of good quality, their size makes it impossible to examine. Break this figure up into at least two figures.*

For the purpose of continuity, as it was the case in the original publication ([Besic et al., 2016]), the entropy scale is defined using "low" and "high" values, which are in the revised version more clearly defined in the text as corresponding to [0,1] range.

Figure 8 was appropriately split in two figures.

**m1:** *Pg 2, line 6: This line is difficult to read, suggest rewrite as: "Hydrometeor classifications are a very popular topic in the weather radar community, particularly since dual polarization radar became a widely used technology (Bringi et al., 2007)." ✓*

**m2:** *Pg 2, lines 9-12: This sentence is long and difficult to read, suggest breaking it up. ✓*

**m3:** *Pg 3, line 29: Phv typically has a value range of 0-1, yet you list -50 - -5.23. Is this value range correct?*

Yes, this range of values refers to the logarithmically transformed $\rho_{hv} \rightarrow \rho'_{hv} = 10 \log (1 - \rho_{hv})$.

**m4:** *Pg 5, line 17 - "...prevents us from applying the..." ✓*

**m5:** *Pg 6, line 7 - "...one can notice the intuitive similarity of the problem." Don't make the assumption that this is as easily viewable by the average reader. Explain why its similar. ✓*

**Page 6: ... : estimating proportions of "pure" components by considering their distances from the observation.**

**m6:** *Pg 6, line 13 - PolSAR, previously it was POLSAR, be consistent with your acronym, also need to define what POLSAR is when it is first used.*

Please refer to #1.m2.

**m7:** *Figure 5: Why is there text in an alternate color? The figure caption should all be a single color of text.*

Indeed, figure captions now all have single text color.

**Page 28: ... (a) the plan with pure hydrometeors (light gray), mixtures (gray), and not very plausible mixtures (dark gray)...**

**m8:** *Figure 8: In addition to the point listed in the major comments, there need to be more tick marks on the altitude axes. There are a number of altitude axes in this figure that don't have any labels at all. This is also true regarding the proportion color bar which shows a scale of 0-100 %. A intermediate value or two is needed here to determine the spacing of the scale. Suggest either 0, 50, 100 % or 0, 25, 50, 75, 100 %.*

Following the suggestion of the reviewer, we facilitated the quantitative interpretation of the presented proportions in Fig. 8 and 9 by introducing additional tick mark labels both on the y-axis and on the colorbar (split Fig. 8 from the original manuscript).

**m9:** *Figure 10: I understand the use of color here, but this should be stated in plain text rather than attempting to use color in the figure caption.*

Please refer to #2.m7.

**m10:** *Figure 11: The labels of the axes are small and difficult to read. Please make these labels larger. This might be a complication of attempting to use 5 plots in a single figure, in which case I suggest breaking the figure up. Additionally, don't use text color in the figure caption, just state the color in words.*

Following the suggestions of the reviewer, we avoided using the text color in the figure captions. However, after enlarging the size of the labels, we found the figure somehow less interpretable, and therefore we decided to keep the current fontsize.

**4   Anonymous reviewer #4**

The paper "Unraveling hydrometeor mixtures in polarimetric radar measurements" introduces classification method which quantifies the dominant contributor to the polarimetric variables in the radar resolution volume and separates it from the secondary contributors. The authors refer to it as bin based de-mixing. The method is novel and merits publication. Nonetheless the paper can be improved and made friendlier to readers. I list in no particular order the more significant issues.

**C1:** *The first part of the paper can stand on its own and I don't see the reason for introducing the Principal Component Analysis, other than to show it is not possible to find linear combinations of the polarimetric variables mutually orthogonal. They state this fact as follows "This implies that PCA cannot be really used as a de-mixing tool, because the coherent backscattering proportion does not correspond to the backscattering of a "pure" hydrometeor type, but rather to the backscattering of the mixture." They also write in the conclusion "there is not a significant rise in incoherency in case of hydrometeor mixtures, on one side strengthens the proposed bin-based approach, and on the other side makes the tools as PCA and ICA less useful in the context of weather radar decomposition/de-mixing." Therefore I suggest they eliminate the section on PCA and ICA. Concentrate on the first part and try to explain better how they do the de-mixing.*

Despite the remark of the reviewer, we decide to keep the second part, assuming that such an approach in assesing the residual incoherency could be very relevant in the weather radar measurements, in particular in case of dealing with a problematic as it is hydrometeor mixing.

**C2:** *I have issues with the terminology "coherency in backscattering" and also with "potential incoherency in the backscattering from different hydrometeors". "Coherent backscattering" is a well-defined and accepted terminology. Check the internet and also look up the articles by Jamison and Kostinski: " Direct Observations of Coherent Backscatter of Radar Waves in Precipitation" and "Partially Coherent Backscatter in Radar Observations of Precipitation" (both in J. At. Science). Here is what they say "The results agree with the earlier conclusions in the previous work, namely that coherent scatter occurs in both rain and snow, that it is larger in snow than it is in rain, and that it can be significant at times." So "coherent backscattering" can occur, it is rare, and not considered in radar meteorology. To be significant the spacing of the drops (hydrometeors) should be comparable and or smaller than the wavelength. For example if the spacing is 1⁄2 of wavelength there would be tremendous enhancement of the returned signal. This is not the case, and I am sure the authors do not mean that it is. They use "coherent" in a different context. So they must devote a paragraph or more to explaining what they mean. After reading the paper I still am not sure? It seems to me they mean the similar type of hydrometeors contribute to the polarimetric variables (like Zh, Zv etc) so that the powers of the backscattered signal from each scatterer add, and*

*the cross product (mutual interactions) cause the variability which is reduce through signal processing.? It this is what they mean they should state so. If I am correct then they could use the term "dominant contribution" for the ones from two or more species, perhaps the dominant can be quantified by %, like if the total of discernible contributions is more than say 60% than these contributors are dominant, the rest is a residue.*

As the reviewer correctly assumed the coherency evoked and analyzed in the paper refers to the invariability of spatial interferences between back and forward scattered waves after their interactions with hydrometeors populating a radar sampling volume. The shuffling achieved by averaging the information over several consecutive pulses, should enforce the hypothesis of coherency in the summing up of individual contributions... Actually, coupled with the hypothesis of relative proximity between the particles, it should enforce the hypothesis of total positive interference (or no interference) meaning that the obtained signal really contains the contributions of all particles present in the radar sampling volume. To resume, the original hypothesis would be that we have incoherent sum of responses of individual particles inside a radar sampling volume, which is "corrected" through the averaging across subsequent pulses by reducing the standard deviation of the distribution of the received power [Sauvageau, 1982].

The coherent scatter observed in the precipitation in [Jameson and Kostinski, 2010b, Jameson and Kostinski, 201 and otherwise associated to the clutter response [Zhang, 2016], basically means that the interferences are not necessarily random and that considering a set of subsequent pulses does not necessarily means the shuffling of precipitating particles. In this case, the averaging does not cancel the interferences (we cannot presume total positive interference), but the residual interferences are not varying in time-space, meaning they are therefore not compromising the assumptions behind the bin-based approach. That is to say, the zero interference scenario is just a special case of the coherent scattering, and both are allowing the parametrized summation of the mixing components. The real adversary would be the incoherency "surviving" the averaging. Why? Because in this case we cannot anymore rely only upon the first order statistics (summation of centroid vectors), but rather have to consider at least the second order statistics (covariance matrices) in dealing with the mixtures.

In this paper, we diagnose the presence of this potentially harmful residual incoherency in the context of mixed radar sampling volumes by involving PCA and ICA applied on the larger neighborhood of contextually similar observations. One could argue that by doing so we are only confirming the validity of the "achieved" coherent scattering (by assuming the proximity of the particles and by averaging over consecutive pulses). However, we do think that this part represent a valuable contribution cause it verifies, in a rather for weather radar community innovative way (PCA and ICA), these assumptions in case of heterogeneousness (mixed) radar sampling volumes, in case where luck of coherency (which is to be compensated through the averaging) is not only due to the intraclass hydrometeor variability but also due to the interclass hydrometeor variability.

The limitation of this technique in detecting the residual incoherency (if the assumption of homogeneity is respected), would be the presence of a very small scale coherency, which could most probably in the spatial context manifest as an evoked residual incoherency.

**Page 17: The second part of the paper is dedicated to the study of a potential influence of the residual spatial incoherency in the backscattering of hydrometeors inside the radar sampling volume. The study is based on adapting the conventional statistical methods, as PCA and ICA, used to deal with the spatial incoherency in the SAR remote sensing, to the specific framework of the weather radar polarimetry. The performance analysis points out the limited influence of the residual incoherency in the regions of hydrometeor mixtures. The introduced evaluation of the spatial consistency in case of heterogeneousness radar sampling volumes is important given that potentially present incoherency in not only due to the intraclass variability but also due to the interclass hydrometeor variability. The**

conclusion, implying that after all there is not a significant rise in incoherency in case of hydrometeor mixtures, on one side strengthens the proposed bin-based approach, and on the other side makes the tools as PCA and ICA less useful in the context of weather radar decomposition/de-mixing than they are in the context of SAR remote sensing.

For the additional changes in the manuscript please refer to #1.C6.

**C3:** *For most part the writing style is but there are quite few awkward spots. Next are some examples and my preferences. Pleas avoid use of targets as these are not missiles, or planes but are hydrometeors or scatterers. Copolar is one word (see the IEEE standards). Check if "hydrometeorly" is an accepted word? The rhv is copolar correlation coefficient (not just correlation). Page 4: By equalizing - should be "by equating". More severely should be "more strictly". This approach "raises" not "rises" (your spell checkers did OK, but no spell checker is yet logical, so let's make one and get rich). Many of your sentences are way too long making these unnecessarily complicated. Example P 14 starting with "This effect, allowing . . .. Has about 58 words.*

We appropriately modified the phrases judged to be awkward by the reviewer.

**C4:** *To be fair to Straka - and true, you should list, " Straka, J.M., 1996: Hydrometeor fields in a supercell storm as deduced from dual-polarization radar. Preprints, 18 Conference on severe Local Storms, San Francisco, AMS. p 551-554. In that paper the fuzzy logic is used, but the weighting functions are pulse like (had top) and are overlapping - which is the essence of fuzziness. And add an appropriate statement - of first use of the technique in pol classifications.*

Indeed, this important contribution is now a part of the references.

**Page 2: In its dominant, supervised form, the methods have been initially based on Boolean logic decision trees [Straka and Zrnic, 1993], before being replaced by a strong tendency to rely on fuzzy logic routine, which is firstly employed by [Straka, 1996], and became a standard tool in the community (e.g. [Vivekanandan et al., 1999]).**

**5 Anonymous reviewer #5**

Nikola Besic et al. This paper presents a technique for performing hydrometeor identification (HID) on polarimetric radar measurements. In particular, it demonstrates a technique to retrieve a mixture of hydrometeors within a single bin, rather than only returning the dominant hydrometeor type. Only being able to return the dominant hydrometeor type has been an Achilles heel of HID. I believe the paper does have a high level of significance. Additionally the figures, other than a few places mentioned below, are of high quality.

**C1:** *The writing is adequate, though some portions would benefit from some editing for clarity. The paper is split into two effective sections. The mathematics in the paper can be confusing to follow as notations and terminology seems to change as the paper progresses. This is caused in part by trying to pull in techniques from several different fields. Overall while I believe the concept has merit, I have some issues with some of the base assumptions. My largest objection is that after identifying the dominant hydrometeor, that the second closest centroid represents the most likely mixture component. I think the radar variables forming correlated frame vectors rather than a true basis make this assumption require quite a bit of justification.*

A very good point! The short part referring to the PolSAR coherent decomposition on standard mechanisms is actually included in order to facilitate explaining the difficulty of not having a true orthogonal basis. In the PolSAR case, we can reasonably assume the orthogonality between the mixing components, which is particularly ensured by projecting the scattering matrix on the orthogonal Pauli matrices. As reviewer correctly stated vectors corresponding to our centroids do not share these properties. Therefore, we're rather influenced by the paradigm using in the hyperspectral unmixing where we also can have a rather correlated frame. This makes a method less analytical and much more empirical, which is why such a large part of the article is dedicated to the experimental verification, rather than on the physical rationalization.

The parts of the paper where we somehow pull in techniques from different fields is slightly reduced in the revised version, hopefully reducing the introduced confusion.

**Page 5: Namely, the coherent decomposition of polarimetric SAR (PoLSAR) data relies on the first-order statistics and represents the scattering matrix of a target as a coherent sum of scattering matrix of elementary interactions (respectively, odd-bounce and double bounce with two different orientations). By comparing this to our polarimetric framework introduced in Sec. 2.1, we can deduce that our standard mechanisms i.e. elementary interactions would correspond to different hydrometeor classes. The important difference which prevents us from applying the equivalent formalism, even under the assumption of the total lack of interferences, is the orthogonality of the basis. That is to say, due to the different physical nature of the elements of the weather radar target vector (Eq. 1), which contains much less geometrical information than the scattering matrix, we cannot expect any orthogonality between the elementary hydrometeors - centroids. Therefore, the starting point of the bin-based de-mixing endeavor is to establish hydrometeor classes as elementary processes and to find a reliable, alternative way to sum them up, which somehow evades the lack of real orthogonal basis.**

**In terms of non-orthogonality, the hyperspectral linear unmixing problem appears more analogous to our polarimetric framework (Fig. 3).**

**C2:** *The comparison of the data with the MASC is a good first step down this path, but as it reports fewer number of hydrometeors (3 ice classified in this case) than the classifier, it makes it hard to know for sure if there is any type of interference that would cause misclassification. I'm not convinced the authors have truly demonstrated that their secondary mode classification works. I would like to see potentially another dataset brought in to show some generality to the method.*

In the very challenging mission of hydrometeor classification validation and particularly in the one concerning the hydrometeor mixtures validation, we tried to do our best. Following the suggestion of the reviewer, the revised version contains an additional analysis involving the MASC based riming degree. This analysis also introduces the temporal scale into the comparison between the radar and the MASC, allowing to see how these two relate along the time periods which are already analyzed in the original version in a rather aggregated way, using the MASC based classification.

**Page 11:The agreement with the MASC is further reinforced by considering the additional parameter estimated from the MASC measurements - the continuous riming degree index (DoR) [Praz et al., 2017]. Independent from the previously mentioned classification, DoR is defined in the range between 0 (no riming) and 1 (graupel). Relying on the previously introduced setup, we now consider only the proportion of rimed ice particles detected by the radar (before and after de-mixing) and the proportion of particles characterized as rimed with different level of strictness with respect to the DoR (DoR$\geq$N, with N being a riming threshold).**

[Figure]

Figure 1: Radar vs. MASC, detection of rimed particles: (a) temporal evolution of the event No. 2, with the shades of gray corresponding to different thresholds (≥) applied on DoR (dark corresponding to 0.5, light to 1), green dashed line being the proportion of RP before the de-mixing and green solid line the proportion of RP after the de-mixing, (b) cross correlation for all events from Table 2, (c) root mean square error (RMSE) for all events from Table 2.

**Page 11**: In Fig. 15a we can see an example (event No. 2) of the temporal evolution of the proportion of the RP as seen by the radar before and after the de-mixing, versus the proportion of rimed particles as seen by the MASC for different thresholds applied on DoR. The moderate temporal correlation which can be intuited from this example is quantified in Fig. 15b for the merged observations from all six events. By introducing the temporal dimension into the analysis we can deduce that the correlation actually slightly decreases after the de-mixing, though not significantly. By checking for the time lagged correlation, we see that the vertical trajectory between the lowest radar sampling volume and the MASC does not seem to compromise significantly the time matching of the samples. Finally, still considering the merged observations, by looking at the RMSE between the estimated proportions for different DoR thresholds in Fig. 15c, one can notice a significant improvement introduced by the de-mixing method.

Fig. 15 in the manuscript corresponds to Fig. 1 in this document.

**C3:** *While my review is somewhat critical, I do think the technique has a fair bit of merit and I would like to see the authors continue it through. In particular, I would love to see a way to address the issue of proportions of secondary components. The authors do equal mixtures for the development of the thresholding of the probability scores, but I believe one could do some simulation to give actual error bars on the retrieval by varying the components of the mixture at different amounts and then performing the retrieval. While real observations always need to be used as a final step, I would have liked to see more simulation data used to characterize the behavior of this retrieval.*

A very useful suggestion! The method is now applied on the synthetic data depicting the particular combinations of interest, with the variable non-equal share of the mixing components.

**Page 8**: Before proceeding to the performance analysis, in order to quantify the potential implicit biases of the introduced parametrization, we apply the de-mixing method on the particular combinations of hydrometeors mixed in both equal and non-equal proportions. Namely, as illustrated in Fig. 8 we consider the mixtures of AG-CR, AG-RP and RN-MH, in the following proportions: 75%-25%, 60%-40%, 50%-50%, 40%-60%, 25%-75%.

**Page 8**: Quantification of the results presented in Fig. 8, provided in Table 1 shows that a mean error for the 50%-50% combination does not exceed 9.2% with a very small standard deviation ($< 1\%$), meaning that we definitely do much better than the classification without de-mixing which in this case can cause a 50% error. A mean error for all combinations does not exceed 12.2% with a very small standard deviation ($< 1\%$), showing again that we necessarily do far better than the classification without the de-mixing.

Table 1: Quantitative evaluation of de-mixing errors (biases) obtained using synthetic (simulated) dataset.

| Mixture | $50\% - 50\%$ $\mu_{error}$ [%] | $50\% - 50\%$ $\sigma_{error}$ [%] | all $\mu_{error}$ [%] | $\sigma_{error}$ [%] |
|---|---|---|---|---|
| AG - CR | 2.5833 | 0.8838 | 12.1988 | 0.9957 |
| AG - RP | 9.1829 | 0.7487 | 9.9326 | 0.7110 |
| RN - MH | 5.9835 | 0.2877 | 11.7355 | 0.1603 |

Fig. 8 in the manuscript corresponds to Fig. 2 in this document. Table 1 in the manuscript corresponds to Table 1 this document.

**M1:** *From the abstract: "Intrinsically related to the concept of entropy, introduced in the context of the radar hydrometeor classification in Besic et al. (2016), the proposed method, based on the hypothesis of coherency in backscattering, estimates the proportions of different 10 hydrometeor types in a given radar sampling volume " As far as I can tell, entropy is not actually used in any of the proposed method, but*

[Figure]

Figure 2: Classification (1) and entropy (2) for different combinations of synthetically produced mixtures: (a) aggregates - crystals, (b) aggregates - rimed ice particles, (c) rain - melting hail.

*rather as a followup parameter to further support the conclusions of mixed hydrometeor types in the bins.*

Entropy is used as an indicator of mixtures and directly depends on the estimates of the proportions.

**M2:** *This paper starts with an analogy to the Cloud and Pottier work on classifying polarimetric SAR, and then attemps to expand this to weather radar using reflectivity, differential reflectivity, Kdp and Rhohv as well as a phase indicator variable. I would like to echo the other commenters that the coherent and incoherent terminology should be explained in text due to its similarity to other common uses of that term in radar.*

Please refer to #1.C6 and #4.C2.

**M3:** *The exposition on page 5 related to POLSAR, while interesting, obscures the discussion of the technique as it applies to weather radar. While entropy is used, the Cloude and Pottier work uses much more than just entropy for their classification, namely adding anisotropy and alpha angle which provide useful information for the POLSAR classification. Those do not necessarily apply here (or at least are not explored in this paper). The switching between different notation and vocabulary can make the work difficult to follow.*

The exposition on page 5 is now reduced, while it still remains true that our concept of entropy is pretty much inspired by the one proposed by Cloude and Pottier, even though their classification effectively uses other parameters derived both from the proportions (anisotropy) and from the parametrization (alpha angle).

**M4:** *I think the approach and workflow could be made much more clear. The classifier listed in this paper is essentially comparing the distance of a measured observation vector to each of the examplars and the minimum distance is assigned to that bin (equation 1). My biggest problem with the work is because the vectors of the space you are embedding into are not orthogonal, you can add 2 hydrometeor types to find a third type. So while I buy that the classifier predicts the dominant type, assuming that the second most likely hydrometeor is present in the mixture feels like a big leap and is not supported in the development of this work. Also what does proportion mean here? I understand it is somewhat of a mathematical construct here, but do the authors feel it is most akin to weight, number, or possibly relative amplitude of the signal returns?*

It is genuinely a big leap, which is nevertheless supported in the pretty elaborated verification part. The proportion refers to the percentage of the radar sampling volume populated by the considered mixing component.

This major remark is pretty much contained in the comment 1 (C1). Please refer to #5.C1.

**M5:** *On page 14 the statement : "As suggested in Subsection 4.1, the first uncorrelated components is supposed to represent a "pure" component, freed up of the effect of incoherence in the data acquisition "*

The idea behind the incoherent decomposition is to treat the direction of the largest variance as the genuine physical information.

**M6:** *Page 6.5-20: There is a lot of switching of terminology here between standard mechanism, end member, centroid, etc.*

Actually, it is true. However, we decided to keep this sort of a comparative introduction, while trying to make the distinction between the quoted terms a bit more clear in the revised version.

**M7:** *Page 6.11-12: "Nevertheless, basing the estimates of proportions on the distance in the space populated by the standard mechanisms and measurements is a reasonable way to sum up the standard mechanisms." I don't feel like this has actually been justified and is a key premise of this work.*

As the reviewer himself correctly remarks, In the absence of the orthogonal framework, the only way to tackle the issue is rather empirical. Therefore, the justification of the quoted premise, which actually reflects one pretty empirical method, can mostly be found in the verification part, now thanks to the reviewer enriched by the additional simulations. Please refer to #5.C1.

**M8:** *"Now that we have identified our centroids as standard mechanisms (analogy with PolSAR) and have found a mode for their addition via the distances in the Euclidean space of the classification " Again, I feel like this is overstated. The reduction in equation 5 and 6 of this to a linear mixing problem does not feel justified. Perhaps this is because $m_i$ is not actually defined here. I assume this is the observations matrix (k) associated with that pure material, but maybe I misread this?*

Vector $m_i$ indeed represents a pure material, and the reduction to the linear mixing problem is, as stated, rather approximative than analytical, as it should be the case for the hyperspectral linear de-mixing also.

**M9:** *"Boxes of hydrometeor mixtures contain different versions 5 of mostly plausible mixtures obtained by linearly combining in equal shares polarimetric parameters of two hydrometeor types involved, following assumptions of coherence and linearity " I would like to see more text as to how this mixing was accomplished. Figure 5 implies you only used equal mixtures of the two hydrometeors (50/50). Why limit it only to equal mixtures? I assume it has to do with finding the threshold value between two centroids that exists at exactly the 50% crossover point, but this should be made more clear.*

Please refer to #5.C3.

**M10:** *Figure 1: I don't understand what the authors are attempting to demonstrate with this figure.*

It would be an effort to illustrate the difference between the framework used in the bin-based de-mixing method and the neighborhood based analysis.

**M11:** *Figure 2: The definition of hydrometeor types in this paper is only in this figure caption. I would add a table or descriptive text with this information.*

With all the added material, we are somehow forced to economize on space by keeping the hydrometeor types in the figure caption and insisting on the provided reference.

**M12:** *Equation 7: This $c_i$ is essentially the same as $m_i$ above correct? Maybe some clarifying statements here, or normalizing the notation would help. In general many of the equations seem to use one set of notation to introduce an equation (I assume the notation from that field), then apply it using a entirely different set of notation. You should unify the different representations.*

Very similar indeed by not really equal. As stated in the manuscript: "However, the major obstacle in applying the very efficient paradigm of the spectral unmixing to the particular context of hydrometeor de-mixing is the fact that our centroids do not really represent the endmembers in the five-dimensional space." This difference justifies the different notations: $\mathbf{m}_i$ for a endmember, for $\mathbf{c}_i$ a centroid.

**M13:** *Page 10: In the comparisons with the MASC, where are the centroids for each HID class drawn from? In particular, are the centroids trained on this dataset, or on another dataset? I'm wondering how much overfitting may be involved in this process that would limit it's generalizability.*

The MASC based hydrometeor classification is based on another dataset, and the overfitting is very much avoided, as it is elaborated in the accompanying reference [Praz et al., 2017]. Though the comment is very much plausible, we would rather insist on reader consulting the reference for such details concerning the employed MASC classification.

**m1:** *Page 3.25 - leptokurticity, while a valid term, will cause many readers to have to stop reading and go look this term up as it is not commonly used in this field. Perhaps add a subclause to describe the term.*

Indeed! The appropriate explanation is added.

**Page 3: ... leptokurticity (fatter distribution tails)...**

**m2:** *Figure 5: It took a few readings to understand that the greyscale level of the boxes and the text in (a) are related. Maybe make this more clear.*

Sorry for the inconvenience! The caption is now changed. Please refer to #3.m7.

**References**

[Balakrishnan and Zrnic, 1990] Balakrishnan, N. and Zrnic, D. S. (1990). Estimation of rain and hail rates in mixed-phase precipitation. *Journal of the Atmospheric Sciences*, 47(5):565–583.

[Bechini and Chandrasekar, 2015] Bechini, R. and Chandrasekar, V. (2015). A semisupervised robust hydrometeor classification method for dual-polarization radar applications. *J. Atmos. Oceanic Technol.*, 32:22–47.

[Besic et al., 2016] Besic, N., Figueras i Ventura, J., Grazioli, J., Gabella, M., Germann, U., , and Berne, A. (2016). Hydrometeor classification through statistical clustering of polarimetric radar measurements: a semi-supervised approach. *Atmospheric Measurement Techniques*, 9:4425–4445.

[Cloude and Pottier, 1996] Cloude, S. R. and Pottier, E. (1996). A review of target decomposition theorems in radar polarimetry. *IEEE Transactions on Geoscience and Remote Sensing*, 34(2):498–518.

[Jameson and Kostinski, 2010a] Jameson, A. R. and Kostinski, A. B. (2010a). Direct observations of coherent backscatter of radar waves in precipitation. *Journal of the Atmospheric Sciences*, 67(9):3000–3005.

[Jameson and Kostinski, 2010b] Jameson, A. R. and Kostinski, A. B. (2010b). Partially coherent backscatter in radar observations of precipitation. *Journal of the Atmospheric Sciences*, 67(6):1928–1946.

[Keat and Westbrook, 2017] Keat, W. J. and Westbrook, C. D. (2017). Revealing layers of pristine oriented crystals embedded within deep ice clouds using differential reflectivity and the copolar correlation coefficient. *Journal of Geophysical Research: Atmospheres*, 122:737–759.

[Praz et al., 2017] Praz, C., Roulet, Y.-A., and Berne, A. (2017). Solid hydrometeor classification and riming degree estimation from pictures collected with a multi-angle snowflake camera. *Atmospheric Measurement Techniques*, 10:1335–1357.

[Sauvageau, 1982] Sauvageau, H. (1982). *Radarmétéorologie*. Éditions Eyrolles, Paris, France.

[Straka, 1996] Straka, J. (1996). Hydrometeor fields in a supercell storm as deduced from dual-polarization radar. In *Preprints, 18th AMS Conference on Severe Local Storms*, San Francisco, CA. Amer. Meteor. Soc.

[Straka and Zrnic, 1993] Straka, J. and Zrnic, D. (1993). An algorithm to deduce hydrometeor types and contents from multi-parameter radar data. In *Preprints, 26th AMS Conf. on Radar Meteorology*, Boston, MA. Amer. Meteor. Soc.

[Tso and Mather, 2009] Tso, B. and Mather, P. (2009). *Classification methods for remotely sensed data*. CRC Press, Boca Raton, FL, 2 edition.

[Vivekanandan et al., 1999] Vivekanandan, J., Zrnic, D. S., Ellis, S. M., Oye, R., Ryzhkov, A. V., and Straka, J. (1999). Cloud microphysics retrieval using s-band dual-polarization radar measurements. *Bull. Am. Meteorol. Soc.*, 80:381–388.

[Wen et al., 2016] Wen, G., Protat, A., May, P. T., Moran, W., and Dixon, M. (2016). A cluster-based method for hydrometeor classification using polarimetric variables. part ii: Classification. *J. Atmos. Oceanic Technol.*, 33:45–60.

[Wen et al., 2015] Wen, G., Protat, A., May, P. T., Wang, X., and Moran, W. (2015). A cluster-based method for hydrometeor classification using polarimetric variables. part i: Interpretation and analysis. *J. Atmos. Oceanic Technol.*, 32:1320–1340.

[Zhang, 2016] Zhang, G. (2016). *Weather Radar Polarimetry*. CRC Press, Inc., Boca Raton, FL, USA, 1st edition.

---

## Author Comment (AC3) · 11 Jul 2018

Please see our responses in the enclosed supplement.

Please also note the supplement to this comment:
https://www.atmos-meas-tech-discuss.net/amt-2018-58/amt-2018-58-AC3-supplement.pdf
* * *